# Induction of lysosomal exocytosis and biogenesis via TRPML1 activation for the treatment of uranium-induced nephrotoxicity

Dengqin Zhong [1,2], Ruiyun Wang [1,2], Hongjing Zhang [1,2], Mengmeng Wang [1], Xuxia Zhang [1] & Honghong Chen [1] ✉

Uranium (U) is a well-known nephrotoxicant which forms precipitates in the lysosomes of renal proximal tubular epithelial cells (PTECs) after U-exposure at a cytotoxic dose. However, the roles of lysosomes in U decorporation and detoxification remain to be elucidated. Mucolipin transient receptor potential channel 1 (TRPML1) is a major lysosomal $Ca^{2+}$ channel regulating lysosomal exocytosis. We herein demonstrate that the delayed administration of the specific TRPML1 agonist ML-SA1 significantly decreases U accumulation in the kidney, mitigates renal proximal tubular injury, increases apical exocytosis of lysosomes and reduces lysosomal membrane permeabilization (LMP) in renal PTECs of male mice with single-dose U poisoning or multiple-dose U exposure. Mechanistic studies reveal that ML-SA1 stimulates intracellular U removal and reduces U-induced LMP and cell death through activating the positive TRPML1-TFEB feedback loop and consequent lysosomal exocytosis and biogenesis in U-loaded PTECs in vitro. Together, our studies demonstrate that TRPML1 activation is an attractive therapeutic strategy for the treatment of U-induced nephrotoxicity.

Uranium (U), a naturally occurring radioactive heavy metal in the Earth's crust, is an important nuclear raw material for nuclear fuel in the nuclear industry. Depleted uranium (DU), a byproduct of nuclear fuel enrichment, also has several civilian and military applications. Both natural U and DU (simply referred as "U" in this paper) are chemical and radioactive toxins with dominant chemical toxicity[1,2]. In cases of accidental exposure or environmental pollution, U can enter the human body by inhaling, ingesting, and wound route and cause health hazards[3–5]. Nephrotoxicity is the hallmark effect of U exposure as the kidney is the main target with U specifically accumulated in renal proximal tubules[3–10]. The therapeutic approach of acute and chronic U intoxication or internal contamination is to accelerate the U excretion from human body by chelating agents, which form non-toxic and soluble complexes with U for eventual renal excretion[11]. However, the therapeutic efficacy of chelating agents is limited for acute exposure with delayed treatment and chronic overexposure[6,12,13].

Intracellular U is present in both soluble and precipitated forms[7,14–16]. An intriguing previous finding is that U in the kidney is selectively concentrated within lysosomes of renal proximal tubular epithelial cells (PTECs) in the form of insoluble phosphate after exposure to soluble U salts in rats[17]. In addition, needle-shape U precipitates with phosphate are present in lysosomes of cultured renal PTECs and many other types of cells after exposure to soluble U salts[15,16,18–25]. The formation and enlargement of U-phosphate precipitates are a U-exposure time- and concentration-dependent dynamic process from soluble form to insoluble isolated thin

[1]Institute of Radiation Medicine, Shanghai Medical College, Fudan University, Shanghai, PR China. [2]These authors contributed equally: Dengqin Zhong, Ruiyun Wang, Hongjing Zhang. ✉e-mail: hhchen@shmu.edu.cn

needles and then to needle-like clusters[15,16,21–24]. Moreover, other toxic heavy-metals such as cerium and niobium are also concentrated and precipitated in the lysosomes of various tissues/cells[25–27]. This intralysosomal precipitation of toxic heavy-metals attributes to the transformed role of the intralysosomal enzyme acid phosphatase[28,29]. Heavy metal-containing precipitates in lysosome are generally considered as an important physiological/biochemical event responsible for heavy-metal detoxification. Therefore, lysosome is an important cellular compartment for removing toxic heavy-metals from the cytoplasm and sequestering them within the vacuolar membrane in an insoluble, detoxified form to limit their diffusion within the organism[25]. However, it remains unclear whether intracellular U and other heavy metals are eliminated via induction of lysosomal exocytosis.

Lysosomal exocytosis is a $Ca^{2+}$-dependent process in which lysosomes fuse directly with the plasma membrane (PM) to release the lysosomal content into the extracellular space in response to an increase of extralysosomal $Ca^{2+}$ [30–35]. The transient receptor potential mucolipin 1 (TRPML1), which is primarily localized in the lysosomal membrane, is a key lysosomal $Ca^{2+}$ release channel[30,36]. Activation of TRPML1 can directly and specifically induce the $Ca^{2+}$-dependent lysosomal exocytosis[37–41]. Transcription factor EB (TFEB) is a transcription factor that mainly regulates both lysosomal biogenesis and autophagy[42–44]. Interestingly, there is a positive feedback loop between TRPML1 and TFEB to promote the lysosomal exocytosis and biogenesis[30,31,45]. While TRPML1 is an essential target gene of TFEB[30,31], TFEB is also activated by lysosomal TRPML1-dependent $Ca^{2+}$ release, inducing the translocation of TFEB from the cytoplasm to the nucleus and consequent transcriptional induction of the coordinated lysosomal expression and regulation (CLEAR) genes and autophagic genes[42–49]. Moreover, TRPML1 is a major effector of TFEB activity. TFEB overexpression reduces the accumulation of lysosomal storage materials via the activation of the lysosomal $Ca^{2+}$ channel TRPML1 and induction of lysosomal exocytosis in cellular and mouse models of lysosomal storage diseases (LSDs)[50]. Accumulating evidence supports an important role of TRPML1 in enhancing lysosomal exocytosis to clear lysosomal lipid storage in cell culture models of LSDs, however, in vivo studies on TRPML1-mediated lysosomal exocytosis and cellular clearance are still lacking[30,31].

In addition to the potential role in releasing lysosomal contents, lysosomal exocytosis is likely involved in the removal of damaged lysosomes as one of endolysosomal damage-response mechanisms[51]. Lysosomal membrane permeabilization (LMP) is characterized by the lysosomal membrane damage, leading to the leakage of the lysosomal acid hydrolases into the cytosol and consequent lysosomal-dependent cell death (LDCD) and other types of cell death such as apoptosis[51–54]. U exposure could induce LMP[55]. Hence, induction of lysosomal exocytosis might contribute to clear the damaged lysosomes and subsequent reduce the cell death induced by U.

In the present study, we evaluated therapeutic effects of the TRPML1 specific agonist ML-SA1 on U-induced nephrotoxicity in mouse models with single- or multiple-dose U exposure. We further studied the mechanisms underlying ML-SA1-mediated protection of U-induced nephrotoxicity in U-loaded renal epithelial cells. Our studies indicate that pharmacological activation of TRPML1 is a promising therapeutic approach for decorporation and detoxification of U-induced nephrotoxicity after acute and chronic U exposure.

## Results and discussion

### The mouse models with single- or multiple-dose U exposure

For acute exposure, mice were intramuscularly injected with a single dose of U at 0.4 mg/kg or 2.0 mg/kg to mimic a toxic dose or highly toxic dose of accidental exposure for 48 h[9]. To mimic chronic exposure, mice were intramuscularly injected with successive 5-day injections of U at 80 μg/kg daily to mimic a low dose environmental exposure for total of 6 days (Supplementary Fig. 1a). Single U exposure

at toxic and highly toxic doses (0.4 mg/kg and 2.0 mg/kg) resulted in a marked increase of U levels in the kidney and 24-h urine (Supplementary Fig. 1b) at 48 h after U injection in a U dose-dependent manner when compared with control mice. U-induced nephrotoxicity is characterized by the induction of tubular lesions in the S3 segment of the proximal tubule[7,56]. Severe tubular necrosis leads to renal dysfunction as reflected by an increased serum creatinine (CRE) and blood urea nitrogen (BUN)[57]. As expected, mice administered with U at the highly toxic dose of 2.0 mg/kg showed severe renal proximal tubular secretion and reabsorption impairments associated with a marked increase of CRE and BUN levels as described by others[9] (Supplementary Fig. 1c). On the other hand, mice injected with U at the toxic dose of 0.4 mg/kg showed no proximal tubular function impairment (Supplementary Fig. 1c), whereas U-induced mild proximal tubular injury was evidenced by the increased levels of kidney injury molecular-1 (KIM-1), a highly sensitive and specific biomarker of proximal tubular damage for early prediction of kidney diseases[58,59], in the S1, S2, and S3 segments of the proximal tubules after single toxic dose U exposure and in the S1 segment of the proximal tubules after single highly toxic dose U exposure (Supplementary Fig. 1d, e). The levels of KIM-1 protein in the S2 and S3 segments were difficult to assess due to serious necrocytosis after single highly toxic dose U exposure at 2.0 mg/kg (Supplementary Fig. 1d). Moreover, the pathological injury in S3 segment with necrotic or exfoliated cells was more profound, which was extended to S2 segment and then S1 segment after single toxic or highly toxic dose U exposure (Supplementary Fig. 1f, g). Similar to the single toxic dose U exposure at 0.4 mg/kg, mice exposed to successive 5-day injections of U at 80 μg/kg daily displayed a marked increase of U levels in the kidney and 24-h urine (Supplementary Fig. 1b) and a mild renal tubular injury with an increase of KIM-1 in the S1, S2, and S3 segments (Supplementary Fig. 1d, e) and increased pathological injury in S3 segment (Supplementary Fig. 1f, g), but no renal function impairment on CRE and BUN levels (Supplementary Fig. 1c) on day 6.

In addition, we performed the lysosomal galectin puncta assay, a highly sensitive method for monitoring early lysosome leakage during LDCD[60], to detect the U-induced LMP in renal PTECs after U exposure. We observed that on day 2 after a single toxic dose U exposure (0.4 mg/kg) and on day 6 after multiple low-dose U exposure (80 μg/kg/day for 5 days), the levels of galectin-1 protein in the S1, S2, and S3 segments of proximal tubules were significantly increased, and there was no significant difference in galectin-1 protein levels between single toxic dose U exposure and multiple exposure of low-dose U (Supplementary Fig. 2a, b). Moreover, the galectin-1 protein level was progressively increased from S1 and S2 segments to S3 segment in both single toxic dose and multiple low-dose U exposures (Supplementary Fig. 2a, b). Notably, on day 2 after highly toxic dose U (2.0 mg/kg) exposure, the level of galectin-1 in the residual intact proximal tubules of the S1 segment was obviously increased (Supplementary Fig. 2a, b). However, galectin-1 was undetectable with immunohistochemical staining due to the large amount of cell death in the S2 and S3 segments (Supplementary Fig. 2a). Consistently, galectin-1 staining clearly demonstrated the puncta formation in renal PTECs of mice exposed to single and multiple doses of U, which was absent in controls, demonstrating the existence of U-induced LMP (Supplementary Fig. 2a, b). Similar results have been reported in renal proximal tubular injury upon oxalate-mediated induction of lysosomal damage[61].

### TRPML1 activation promoted the removal of U accumulated in the kidney of mice after single- or multiple-dose U exposure

ML-SA1 is a potent membrane-permeable specific TRPML1 small molecule agonist[62] and can effectively promote the lysosomal cholesterol clearance and reduce the accumulation of intracellular misfolded protein α-synuclein through triggering lysosomal exocytosis in cellular models of LSDs[62] and neurodegenerative disease[63], although this action has not been established in vivo yet. We investigated the

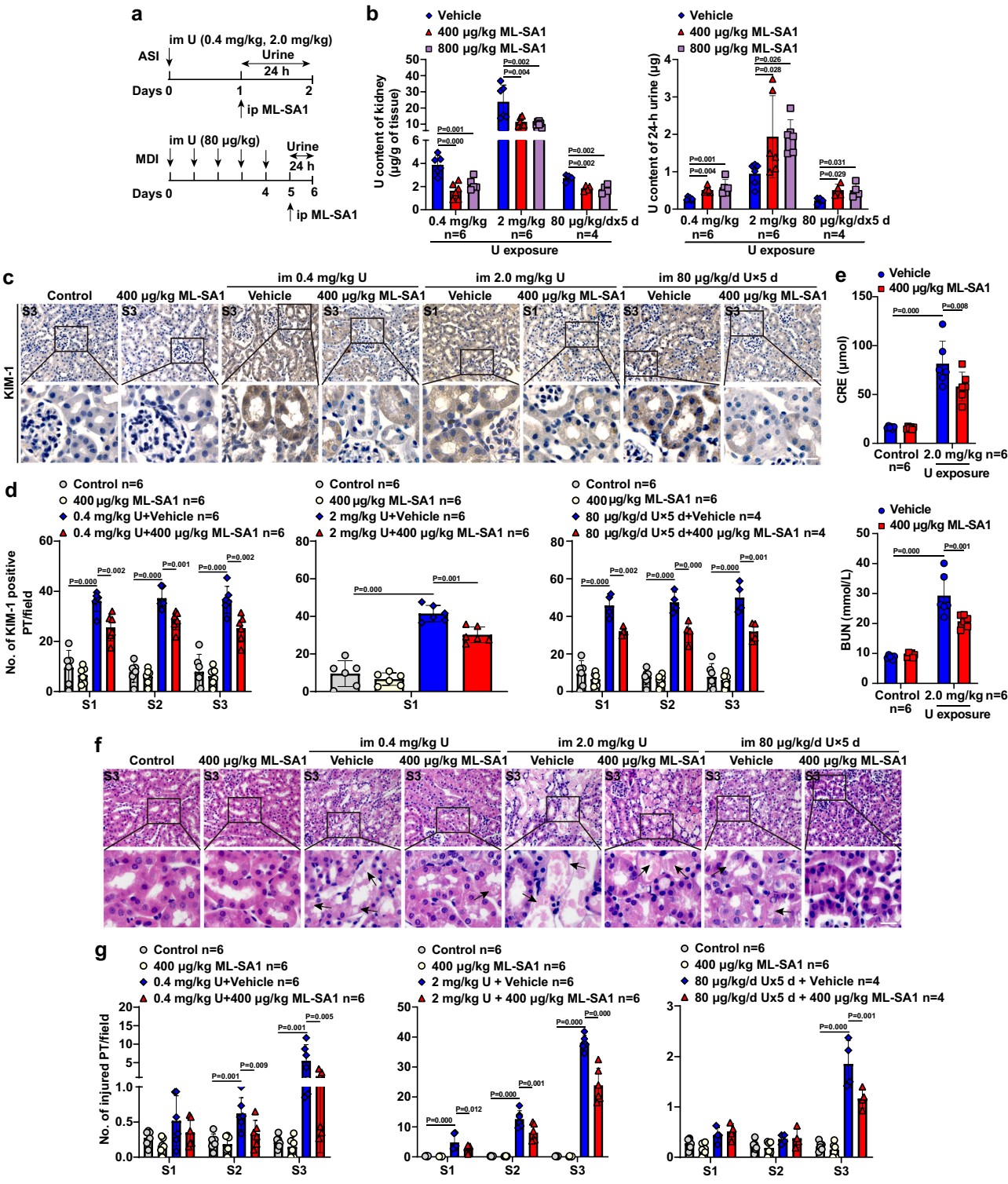

effects of ML-SA1 on urinary U excretion and U accumulation in the kidney of mice after U exposure (Fig. 1a). Twenty-four hour-delayed administration of ML-SA1 at 400 or 800 µg/kg not only significantly lowered U content in kidney to 42–57% but also increased the 24-h urinary U excretion to 186–213% in mice with a single toxic dose U at 0.4 mg/kg or highly toxic dose U at 2.0 mg/kg (Fig. 1b), demonstrating that ML-SA1 has pronounced effects on the removal of U in the kidney even under the condition with the severe functional damage and pathological injury of proximal tubules (Supplementary Fig. 1c, f, g). The therapeutic effect of ML-SA1 may attribute to its action before U-induced cell death even in the presence of U-induced partial and

selective LMP (Supplementary Fig. 2a, b). Moreover, 5 day-delayed ML-SA1 treatment at 400 and 800 µg/kg after multiple doses of U exposure also significantly decreased the U content in kidney to 65–67% and increased the 24-h urine U excretion to 217–218% (Fig. 1b), demonstrating that ML-SA1 effectively decreases the intracellular U accumulation in the kidney after repeated U exposure.

At present, sodium bicarbonate is the only drug for U removal in clinical use, which is effective for the treatment of acute U poisoning when the urine reaches a pH of eight to nine in affected humans[11,64]. We found that immediate administration of sodium bicarbonate at 1.0 g/kg induced urine alkalinization (pH 8 to 9), markedly decreased the U

**Fig. 1 | TRPML1 agonist ML-SA1 promotes renal U clearance and reduces the U-induced renal proximal tubule injury in mice after single- or multiple-dose U exposure. a** Schematic representation of the experiments on single- or multiple-dose U exposure followed by ML-SA1 treatment at 400 or 800 μg/kg in BALB/c male mice. $n$ = 6 mice for control groups, ML-SA1 treatment alone groups, single dose U exposure alone groups and signal dose U exposure followed by ML-SA1 treatment groups, and $n$ = 4 mice for multiple-dose U exposure alone group and multiple-dose U exposure followed by ML-SA1 treatment group. **b** U contents in kidney tissues and the last 24-h urine from mice with single- or multiple-dose U exposure and ML-SA1 treatment. **c** Representative immunohistochemical staining of KIM-1 in the S3 or S1 segment of proximal tubules in renal cortex of mice after U exposure and ML-SA1 treatment as indicated in the figure. Boxed areas are enlarged below. **d** Quantitative analysis of KIM-1 levels in the S1, S2, and S3 segments of proximal tubules as shown in **c** and Supplementary Fig. 4a. **e** The CRE and BUN levels in mice after single-dose (2.0 mg/kg) U exposure and treatment with ML-SA1 at 400 μg/kg or vehicle. **f** Representative H&E staining in the S3 segment of proximal tubules in renal cortex of mice after U exposure and ML-SA1 treatment as indicated in the figure. Boxed areas are enlarged below. **g** Quantitative analysis of pathological injury of proximal tubules with necrotic or exfoliated cells in the S1, S2, and S3 segments of proximal tubules as shown in **f** and Supplementary Fig. 4b. im: intramuscular injection; ip: intraperitoneal injection; ASI: a single injection; MDI: multiple dose injection; PT: proximal tubule. Data represent mean ± SD. Statistical significance was evaluated by one-way ANOVA with LSD's post hoc test (**b, d, e, g**). Source data are provided as a Source Data file. All the images share the same scale bar (20 μm).

content in the kidney and increased the urinary U excretion in 24-h urine in mice with U exposure at a single high toxic dose U at 2.0 mg/kg (Supplementary Fig. 3a, b), which is consistent with the studies on U decorporation by sodium bicarbonate in U-contaminated rats[65]. However, the 24 h-delayed and 5 day-delayed single-dose administration of sodium bicarbonate had no effects on U decorporation in mice with U exposure at a single highly toxic dose U at 2.0 mg/kg or successive 5-day injections of low dose U at 80 μg/kg daily (Supplementary Fig. 3a, b). Recently, it has been reported that 24 h-delayed multiple-dose administration of chelating agent 5LIO-1-Cm-3,2-HOPO induced a 61.4–65.0% U reduction in kidneys of mice after an intravenous injection of U (0.5 mg/kg)[66]. Evidently, the 24 h-delayed single-dose administration of ML-SA1 is as effective as the 24 h-delayed multiple-dose administration of 5LIO-1-Cm-3,2-HOPO in decorporation efficiency. More importantly, even the 5 day-delayed single-dose administration of ML-SA1 had significant effects on removing U from the kidney and enhancing the urinary U excretion in mice after multiple doses of U exposure, although the removal efficiency of U in the kidney is lower when compared to the 24 h-delayed administration of ML-SA1.

## TRPML1 activation mitigated the U-induced renal proximal tubular injury in mice after single- or multiple-dose U exposure

As the removal efficiencies of ML-SA1 at 400 and 800 μg/kg were not significantly different from each other (Fig. 1b), we selected the dose at 400 μg/kg of ML-SA1 to perform the follow-up experiments. Twenty-four h-delayed ML-SA1 treatment not only significantly decreased the level of KIM-1 in S1, S2 and S3 segments and alleviated the renal pathological injuries with necrosis or desquamation of PTECs in S2 and S3 segments of the proximal tubule at 48 h after exposure to the single toxic dose of U at 0.4 mg/kg (Fig. 1c, d, f, g and Supplementary Fig. 4a, b), but also attenuated the renal proximal tubular function impairment with significant decreases of CRE and BUN levels and attenuation of the renal pathological injuries in S1, S2 and S3 segments of the proximal tubule at 48 h after the single high-toxic-dose U intoxication at 2.0 mg/kg (Fig. 1e–g and Supplementary Fig. 4b). Similarly, 5 day-delayed treatment with ML-SA1 at 400 μg/kg significantly reduced the levels of KIM-1 in S1, S2 and S3 segments and attenuated the renal pathological injuries in S3 segment of the proximal tubule on day 6 after multiple low-dose U exposure (80 μg/kg/day for 5 days) (Fig. 1c, d, f, g and Supplementary Fig. 4a, b). The amelioration effect of ML-SA1 on U-induced proximal tubular injuries was associated with the significant reduction of the renal U concentrations by ML-SA1 in mice with a single high dose U at 0.4 mg/kg or 2.0 mg/kg or successive 5-day injections of low dose U at 80 μg/kg daily (Fig. 1b). With the reduction of U content in the kidney (Supplementary Fig. 3b), immediate administration of sodium bicarbonate ameliorated the renal proximal tubular function impairment with significant decreases of CRE and BUN levels and a marked reduction of the renal pathological injuries in S1, S2 and S3 segments of the proximal tubule at 48 h after the single high-toxic-dose U intoxication at 2.0 mg/kg (Supplementary Fig. 3c–e). In contrast, the 24 h-delayed and 5-day-delayed single-dose administration of sodium bicarbonate

were ineffective in alleviating renal tubular injury (Supplementary Fig. 3c–e), which was consistent with the results of ineffectiveness on U content in the kidney and urinary U excretion (Supplementary Fig. 3b) in mice with a single highly toxic dose U at 2.0 mg/kg or successive 5-day injections of low dose U at 80 μg/kg daily. Altogether, these findings demonstrate that ML-SA1 treatment is a potent medical countermeasure for delayed administration under various degrees of renal tubular injury conditions after single or multiple U exposure.

## TRPML1 activation increased apical exocytosis of lysosomes in renal PTECs of mice after single- or multiple-dose U exposure

TRPML1 is a key regulator of lysosomal exocytosis[37–41]. Lysosomal-associated membrane proteins LAMP-1 and LAMP-2 are two major lysosomal membrane proteins[30,44]. The translocation of LAMP-1/LAMP-2 and lysosomal ion channel TRPML1 to the PM is the hallmark of lysosomal exocytosis[31,34,38,50,67]. Polarized exocytosis of lysosomes can be monitored by the asymmetric appearance of LAMP-1 or LAMP-2 on the PM in polarized epithelial cells[68,69]. Studies have revealed that before lysosomes fuse with PM, lysosomes translocate from perinuclear zone to the region adjacent to PM[30,33,50,67]. We found that LAMP-1 as well as TRPML1 was localized to the apical membrane, but not to the basolateral membrane, of S1, S2, and S3 segments of proximal tubules of kidney in control mice (Fig. 2a, b and Supplementary Fig. 5), indicating that lysosomal exocytosis directed toward the proximal tubular lumen is active at the normal renal physiological conditions. ML-SA1 treatment alone (400 μg/kg) significantly induced LAMP-1 and TRPML1 localization to the apical membrane in S1, S2, and S3 segments of proximal tubules (Fig. 2a, b and Supplementary Fig. 5), suggesting that apical exocytosis of lysosomes is effectively activated. Notably, both the single toxic dose (0.4 mg/kg) and multiple low-dose (80 μg/kg/day for 5 day) of U exposures significantly increased the levels of LAMP-1 and TRPML1 proteins in the apical membrane of S1, S2, and S3 segments of proximal tubules compared with controls. While the levels of LAMP-1 and TRPML1 proteins in the S2 and S3 segments were difficult to assess due to serious necrocytosis (Supplementary Fig. 5), they were significantly increased in the apical membrane of intact proximal tubules of S1 segment in mice exposed to the single highly toxic dose of U (2.0 mg/kg) (Fig. 2a, b), indicating that U exposure alone also activates the apical exocytosis of lysosomes in PTECs, which reflects the self-detoxification in PTECs. Importantly, treatment with ML-SA1 at 400 μg/kg further markedly increased the apical membrane staining levels of LAMP-1 and TRPML1 in S1, S2, and S3 segments of proximal tubules in the mice exposed to the single (0.4 mg/kg) or multiple low-dose (80 μg/kg/day for 5 days) U and in S1 segment of proximal tubules in mice exposed to the single highly toxic dose of U (2.0 mg/kg) compared with the corresponding U exposure plus vehicle group (Fig. 2a, b and Supplementary Fig. 5), suggesting that ML-SA1 promotes the apical exocytosis of lysosomes in renal PTECs of mice exposed to U. The absence of dose-response effect of U removal by ML-SA1 at 400 and 800 μg/kg (Fig. 1b) may be due to a limitation on the number of lysosomes in which exocytosis occurs.

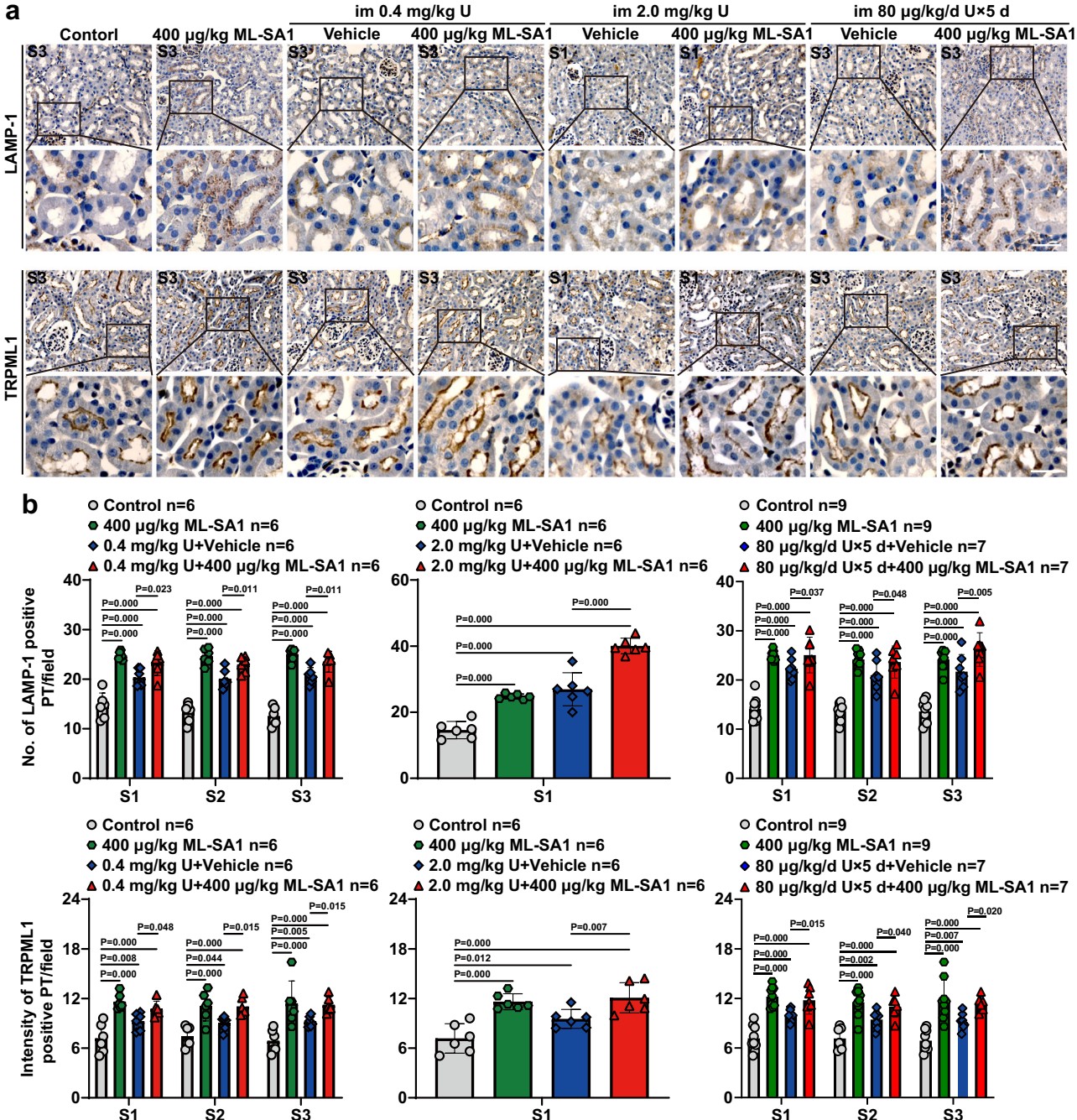

**Fig. 2 | TRPML1 agonist ML-SA1 increases the levels of lysosomal membrane proteins in the apical membrane of renal proximal tubules in mice after single- or multiple-dose U exposure.** The male mice with single- or multiple-dose U exposure and ML-SA1 treatment were described in Fig. 1a. $n = 6$ mice for single-dose U exposure alone groups, single-dose U exposure followed by ML-SA1 treatment groups, and corresponding control group and ML-SA1 treatment alone group. $n = 7$ mice for multiple-dose U exposure alone group and multiple-dose U exposure followed by ML-SA1 treatment group, and $n = 9$ mice for corresponding control group and ML-SA1 treatment alone group. **a** Representative immunohistochemical staining of LAMP-1 and TRPML1 in the S3 or S1 segment of the proximal tubules in renal cortex of mice after U exposure and ML-SA1 treatment as indicated in the figure. Boxed areas are enlarged below. Images share the same scale bar (20 μm). **b** Quantitative analysis of LAMP-1 and TRPML1 staining of the apical membrane in the S1, S2, and S3 segments of renal proximal tubules as shown in **a** and Supplementary Fig. 5. PT: proximal tubule. Data represent mean ± SD. Statistical significance was evaluated by one-way ANOVA with LSD's post hoc test (**b**). Source data are provided as a Source Data file.

## TRPML1 activation decreased U-induced LMP and apoptosis in renal PTECs of mice after single- or multiple-dose U exposure

ML-SA1 treatment significantly decreased the levels of galectin-1 in the PTECs of the S2 and/or S3 segments of proximal tubules in mice exposed to the single toxic dose (0.4 mg/kg) or multiple low-dose (80 μg/kg/day for 5 days) of U and in the PTECs of the S1 segment of proximal tubules in mice exposed to the single highly toxic dose of U

(2.0 mg/kg) (Fig. 3a, b and Supplementary Fig. 6a), suggesting that ML-SA1 decreases the accumulation of lysosomes with LMP or mitigates U-induced LMP. This effect of ML-SA1 may be related to endo-lysosomal damage-response mechanism that involves the removal of damaged and dysfunctional lysosomes by exocytosis[51].

LMP and the subsequent leakage of lysosomal hydrolases into the cytosol can lead to LDCD with apoptotic, apoptosis-like or necrotic

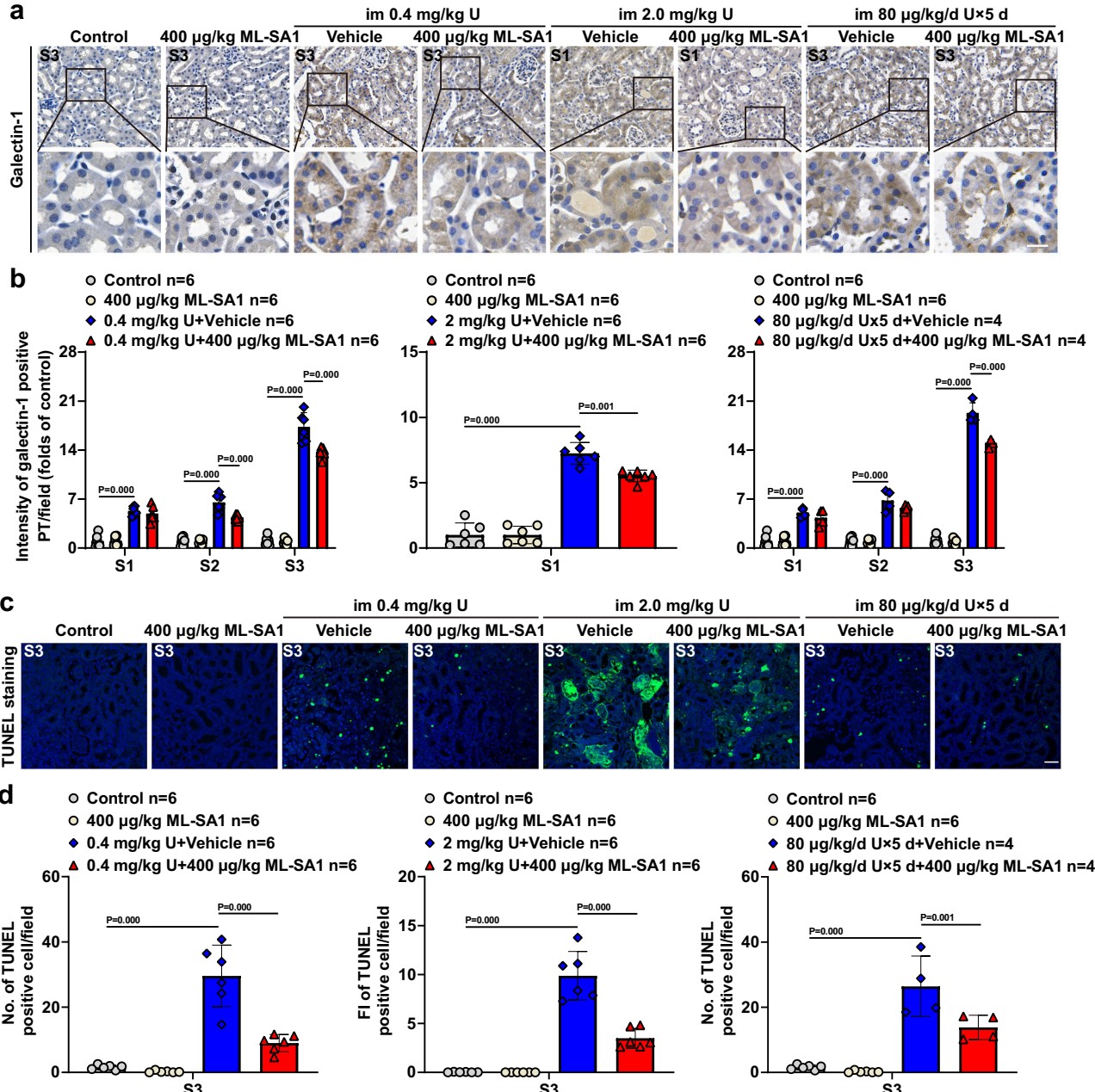

**Fig. 3 | TRPML1 agonist ML-SA1 reduces the U-induced LMP and LMP-related apoptosis of renal PTECs in mice after single- and multiple-dose U exposure.** The male mice with single- or multiple-dose U exposure and ML-SA1 treatment were described in Fig. 1a. *n* = 6 mice for control groups, ML-SA1 treatment alone groups, single dose U exposure alone groups and signal dose U exposure followed by ML-SA1 treatment groups. *n* = 4 mice for multiple-dose U exposure alone group and multiple-dose U exposure followed by ML-SA1 treatment group. **a** Representative immunohistochemical staining of galectin-1 in the S3 or S1 segment of the proximal tubules in renal cortex of mice after U exposure and ML-SA1 treatment as indicated in the figure. Boxed areas are enlarged below. **b** Quantitative analysis of galectin-1 staining in the S1, S2, and S3 segments of the proximal tubules as shown in **a** and Supplementary Fig. 6a. **c** Representative TUNEL staining in the S3 segment of the proximal tubules in renal cortex of mice after U exposure and ML-SA1 treatment as indicated in the figure. **d** Quantitative analysis of TUNEL staining in the S3 segment of the proximal tubules as shown in **c**. PT: proximal tubule; FI: fluorescence intensity. All the images share the same scale bar (20 μm). Data represent mean ± SD. Statistical significance was evaluated by one-way ANOVA with LSD's post hoc test (**b**, **d**). Source data are provided as a Source Data file.

features and apoptosis[51,52,70]. TUNEL staining showed that apoptotic cells were mostly found in S3 segment of proximal tubules when mice were exposed with the toxic doses of U (the single dose at 0.4 mg/kg or multiple doses at 80 μg/kg/day for 5 days), and that extensive apoptosis was found in the S3 segment of proximal tubules when mice were exposed to the highly toxic dose of U (single dose of 2.0 mg/kg) (Fig. 3c, d and Supplementary Fig. 6b). Consistent with the results on galectin-1, 24 h-delayed ML-SA1 treatment significantly attenuated U-induced apoptosis in the S3 segment of renal proximal tubules in

mice exposed to U with the single toxic or highly toxic dose (0.4 mg/kg or 2.0 mg/kg) or multiple low-dose (80 μg/kg/day for 5 days) (Fig. 3c, d and Supplementary Fig. 6b). These results indicate that ML-SA1 reduces the U-induced LMP and apoptosis through the removal of damaged lysosomes by exocytosis.

### U-loaded cellular models with short-term U exposure
To further dissect the mechanism underlying TRPML1 activation-mediated protection against U-induced nephrotoxicity, we established

U-loaded cellular models with short-term U exposure. We exposed renal epithelial HK-2 cells to U at different concentrations for 24 h to induce cellular damage at different levels after short-term U exposure. U-induced cytotoxicity was evaluated by lactate dehydrogenase (LDH) release assay[71], showing a dose-related cytotoxic response with U concentrations ranging from 100 μM to 1200 μM. The half maximal inhibitory concentration ($IC_{50}$) occurred at 665 μM after 24 h-U exposure (Supplementary Fig. 7a), which was similar to that reported in the previous studies[24,72,73]. In addition, U exposure at 50, 100, and 600 μM for 24 h led to a marked increase in intracellular U content (Supplementary Fig. 7b). Furthermore, there was a significant upregulation of KIM-1 and liver-type fatty acid-binding proteins (L-FABPs), two high sensitive and specific biomarkers of proximal tubule damage[58,74,75], in HK-2 cells exposed to 100 and 600 μM U for 24 h, but not 50 μM U for 24 h (Supplementary Fig. 7c–f), confirming that U concentrations of 50, 100, and 600 μM for 24 h exposure are non-cytotoxic, sub-cytotoxic and cytotoxic, respectively. Finally, LMP in U-loaded HK-2 cells was detected by galectin immunofluorescence staining[60]. To ensure that galectin-3 labeled damaged lysosomes specifically, we examined its colocalization with lysosomal membrane protein LAMP-1. Exposure to U markedly increased the colocalization of galectin-3 and LAMP-1 in HK-2 cells exposed to 100 and 600 μM for 24 h (Supplementary Fig. 7g, h), suggesting that U induces LMP in a U concentration-dependent manner.

**TRPML1 activation enhanced the removal of intracellular U and mitigated U-induced LMP and cell death through promoting the lysosomal exocytosis in U-loaded HK-2 cells after short-term U exposure**

HK-2 cells were exposed to U for 24 h followed by ML-SA1 treatment for 30 min (Fig. 4a). Twenty-four h-delayed ML-SA1 treatment resulted in an increase of the exposure of the LAMP-1 luminal domain on the PM in both control cells and U-loaded HK-2 cells at 30 min after 24 h-U exposure at 600 μM (Fig. 4b). Meanwhile, TRPML1 was detected around the cells, suggesting a PM localization of the protein (Fig. 4b). Consistently, significantly higher levels of lysosomal hydrolytic enzyme β-hex, a direct consequence of lysosomal exocytosis[67], were detected in the culture media of both control cells and U-loaded HK-2 cells treated with ML-SA1 for 30 min after 24 h-U exposure at 50, 100, and 600 μM (Fig. 4c), suggesting that ML-SA1 triggers the lysosomal exocytosis after exposure to U at non-toxic, sub-toxic and even toxic concentration.

Vacuolin-1 is a potent blocker of $Ca^{2+}$-dependent lysosomal exocytosis[76]. Treatment with vacuolin-1 alone significantly reduced the levels of β-hex in the culture media of control cells and U-loaded HK-2 cells at 30 min after 24 h-U exposure at U concentrations of 50, 100, and 600 μM (Fig. 4c). Importantly, ML-SA1-induced β-hex release was dramatically attenuated by Vacuolin-1 in U-loaded HK-2 cells at 30 min after 24 h-U exposure (Fig. 4c). These results further demonstrate that ML-SA1 has the potent activity in promoting the lysosomal exocytosis in U-loaded HK-2 cells even with U exposure at toxic concentration.

Meanwhile, 24 h-delayed ML-SA1 treatment significantly reduced the intracellular U content and increased extracellular U content in the culture medium of U-loaded HK-2 cells at 30 min after 24 h-U exposure at 50, 100, and 600 μM (Fig. 4d), and vacuolin-1 displayed opposite effects (Fig. 4d). Notably, the effect of ML-SA1 on the intracellular U depletion was remarkably blocked by Vacuolin-1 (Fig. 4d), suggesting that ML-SA1 treatment can effectively remove the intracellular U by promoting the lysosomal exocytosis after exposure to U at non-toxic, sub-toxic and even toxic concentrations.

Furthermore, 24 h-delayed ML-SA1 treatment significantly reduced U-induced LMP as measured by galectin-3 and LAMP-1 colocalization, and Vacuolin-1 displayed opposite effects in U-loaded HK-2 cells exposed to U at 100 and 600 μM for 24 h (Fig. 5a, d). Importantly, the effect of ML-SA1 on attenuation of the U-induced LMP was

abolished by Vacuolin-1 (Fig. 5a, d), suggesting that ML-SA1 stimulates exocytosis to enhance the elimination of damaged lysosomes with LMP. In addition, the U-induced mortality and level of cleaved caspase-3 protein were significantly decreased by ML-SA1 and enhanced by Vacuolin-1 treatment in U-loaded HK-2 cells at 30 min after 24 h-U exposure at 100 and 600 μM (Fig. 5b, c, e, f). Consistent with the results on intracellular U clearance (Fig. 4d), the effects of ML-SA1 on reduction of the U-induced cell death and apoptosis was abolished by Vacuolin-1 (Fig. 5b, c, e, f). Therefore, the removal of damaged lysosomes by exocytosis might prevent the release of lysosomal proteolytic enzymes (i.e. cathepsins) into the cytoplasm and consequent cell death. Together, these findings demonstrate that ML-SA1-induced lysosomal exocytosis have dual protective effects on expelling cytotoxic U from the cells and preventing LMP-mediated cell death.

**TRPML1 activation induced the formation of a positive TRPML1-TFEB feedback loop for the activation of lysosome biogenesis and exocytosis in U-loaded HK-2 cells after short-term U exposure**

To define the mechanisms by which ML-SA1 promotes the removal of intracellular U via lysosomal exocytosis, we examined the effects of ML-SA1 on the TRPML1-TFEB pathway in U-loaded HK-2 cells after short-term U exposure. Lysosomal $Ca^{2+}$ release via TRPML1 is known to induce TFEB dephosphorylation and subsequent nuclear translocation through the activation of the phosphatase calcineurin in response to starvation[49]. Thus, we first tested whether ML-SA1 triggered the $Ca^{2+}$ efflux from lysosomes in U-loaded HK-2 cells. Consistent with the previous work on ML-SA1 for cytosolic $Ca^{2+}$ induction in HEK293T cells expressing mutant TRPML1 channels[62], we found that ML-SA1 treatment resulted in an obvious cytosolic $Ca^{2+}$ elevation in both control and U-loaded HK-2 cells with 24 h-U exposure at 100 and 600 μM after pretreatment with ionomycin, a $Ca^{2+}$ ionophore, to induce $Ca^{2+}$ release from all intracellular stores apart from lysosomes[77,78] (Fig. 6a, b). Moreover, lysosomal $Ca^{2+}$ release induced by ML-SA1 in U-loaded cells with 24 h-U exposure at 100 and 600 μM was significantly higher than that in the control cells (Fig. 6a, b). As expected, 24-h delayed ML-SA1 treatment for 30 min significantly increased the dephosphorylation of TFEB (downshift of TFEB molecular weight), nuclear translocation of TFEB and levels of the TFEB downstream targets LAMP-1 and TRPML1 proteins in both control and U-loaded HK-2 cells with 24-h exposure at 100 and 600 μM (Fig. 6c–h). Moreover, ML-SA1-induced TFEB nuclear translocation in both control and U-loaded HK-2 cells was abolished by shRNA-mediated depletion of TRPML1 and siRNA-mediated depletion of PPP3CB, one of the subunits of calcineurin (Fig. 6i, j, Supplementary Fig. 8a–f), suggesting that ML-SA1 activates the TRPML1-TFEB pathway and induces lysosomal biogenesis in U-loaded HK-2 cells after exposure to U even at the toxic concentration. Following the 'omnis membrana e membrana' (membranes can be generated only from membranes) dogma[79], lysosomal exocytosis and biogenesis are in dynamic balance, which is required to maintain the cellular lysosomal capacity and lysosomal activity for cell survival[80]. We thus propose that lysosomal biogenesis activated by ML-SA1 treatment is a compensatory mechanism to generate more new lysosomes to compensate for the lysosomal consumption due to increased exocytosis. Moreover, compensatory lysosomal biogenesis by ML-SA1 might also contribute to enrichment of intracellular U in lysosomes.

Notably, 24 h-U exposure at 100 and 600 μM alone also led to significant increases in TFEB dephosphorylation, TFEB nuclear translocation, and LAMP-1 and TRPML1 protein levels in a U exposure concentration-dependent manner (Fig. 6c–h). Moreover, U-induced TFEB nuclear translocation was impaired in the HK-2 cells with knockdown of TRPML1 and PPP3CB, but still obviously higher than that of vector- or scramble control siRNA-transfected cells (Fig. 6i, j, Supplementary Fig. 8a–f), suggesting that U-induced TFEB nuclear translocation was partly through TRPML1 and calcineurin

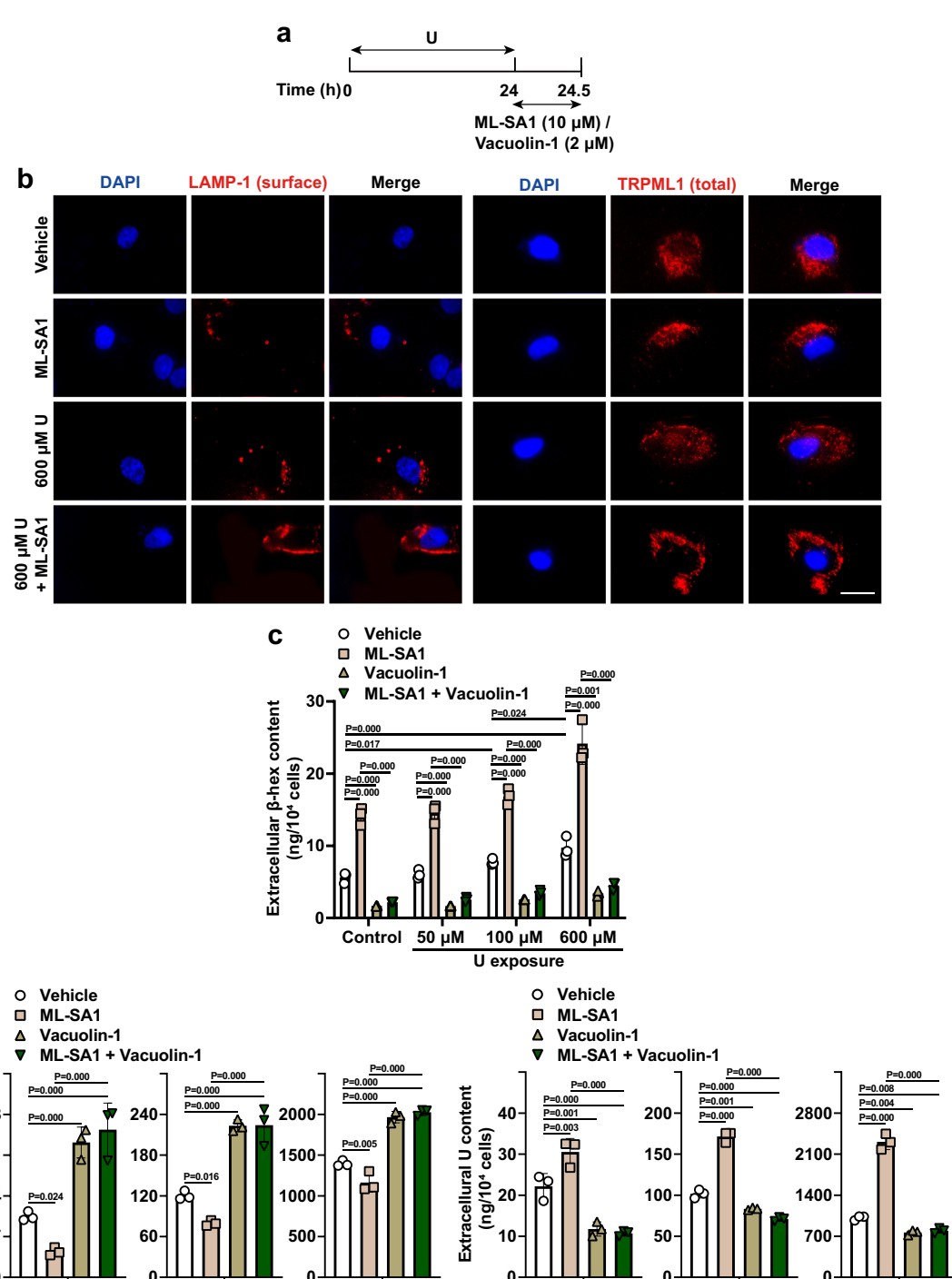

**Fig. 4 | TRPML1 activation with ML-SA1 triggers an efficient removal of intracellular U by lysosomal exocytosis in U-loaded renal epithelial HK-2 cells. a** The timeline of short-term U (50, 100, 600 μM) exposure for 24 h followed by ML-SA1 (10 μM) and Vacuolin-1 (2 μM) treatment for 30 min after washout of U. **b** Representative fluorescence images showing the LAMP-1 exposure on the PM in nonpermeabilized HK-2 cells and TRPML1 localization in permeabilized HK-2 cells after U exposure and ML-SA1 treatment. Images share the same scale bar (20 μm). **c** Comparison of β-hex release in HK-2 cells after U exposure and ML-SA1/Vacuolin-1 treatment. **d** Comparison of intracellular and extracellular U contents in HK-2 cells after U exposure and ML-SA1/Vacuolin-1 treatment. PM: plasma membrane. Data represent mean ± SD. $n = 3$ independent experiments. Statistical significance was evaluated by one-way ANOVA with LSD's post hoc test (**c**, **d**). Source data are provided as a Source Data file.

and additional mechanisms may be involved. Although 24 h-U exposure at 100 and 600 μM alone did not induce changes in the overall levels of cytosolic Ca²⁺ (Fig. 6a), treatment with the Ca²⁺-chelator BAPTA-AM inhibited U-induced TFEB nuclear translocation (Supplementary Fig. 9a, b), suggesting that 24 h-U exposure may

trigger a sustained local calcium signal, leading to Ca²⁺-dependent lysosomal exocytosis showing the enrichment of LAMP-1 on the PM and the increase of β-hex level in the culture medium of U-loaded HK-2 cells at 30 min after 24 h-U exposure at U concentrations of 100 and 600 μM (Fig. 4b, c). Based on this observation and the

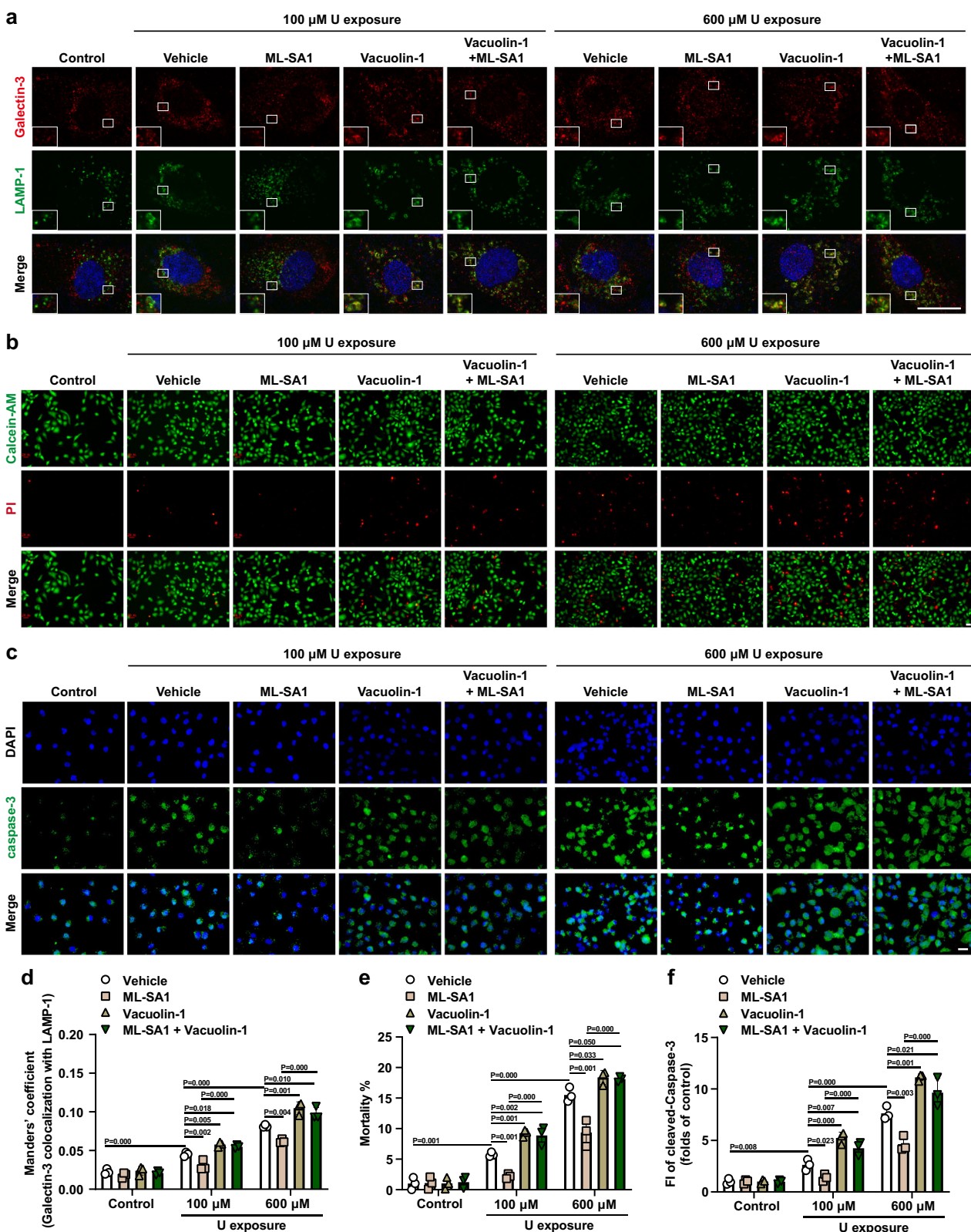

finding on partial LMP caused by 24 h-U exposure at 100 and 600 µM (Fig. 5a, d), we questioned whether 24 h-U exposure affected the levels of lysosomal $Ca^{2+}$. The intra-luminal $Ca^{2+}$ levels in the lysosomes had no marked differences between control and U-loaded cells although U-loaded HK-2 cells with 24 h-U exposure at 600 µM showed a slightly decrease (Supplementary Fig. 9c, d), suggesting that there may be some compensatory mechanisms to

maintain $Ca^{2+}$ homeostasis in lysosomes[77]. Studies have revealed that TFEB nucleo-cytoplasmic shuttling is mainly negatively modulated by mTORC1 (mechanistic target of rapamycin complex 1)-mediated TFEB phosphorylation via a substrate-selective mechanism that relies on the activity of RagC/D GTPases in the recruitment of mTORC1 and TFEB to the lysosomal surface[81–88]. Moreover, it has been reported that lysosomal damage selectively

**Fig. 5 | TRPML1 activation with ML-SA1 reduces the U-induced LMP and LMP-related cell death via lysosomal exocytosis in U-loaded renal epithelial HK-2 cells.** The HK-2 cells were incubated with U at 0, 100, 600 μM for 24 h. After washout of U, the cells were treated with vehicle, ML-SA1 (10 μM), Vacuolin-1 (2 μM), and ML-SA1 (10 μM) plus Vacuolin-1 (2 μM) for 30 min and then analyzed. **a** Representative fluorescence images for colocalization of galectin-3 (red) and LAMP-1 (green) in HK-2 cells after U exposure and ML-SA1/Vacuolin-1 treatment as indicated in the figure. **b** Representative fluorescence images of Calcein-AM/PI staining in HK-2 cells after U exposure and ML-SA1/Vacuolin-1 treatment as indicated in the figure. The living cells were stained with Calcein-AM (green), and the nuclei of the dead cells were stained with PI (red). **c** Representative images of immunofluorescence staining of cleaved-caspase-3 (green) in HK-2 cells after U exposure and ML-SA1/Vacuolin-1 treatment as indicated in the figure. **d** Quantitative analysis of colocalization of galectin-3 with LAMP-1 under various treatment conditions shown in **a**. More than 30 cells were analyzed in each sample. **e** Quantitative analysis of the cell mortality under various treatment conditions shown in **b**. About 1000 cells were analyzed in each sample. **f** Quantitative analysis of cleaved-caspase-3 level under various treatment conditions shown in **c**. More than 500 cells were analyzed in each sample. FI: fluorescence intensity. All the images share the same scale bar (20 μm). Data represent mean ± SD. $n = 3$ independent experiments. Statistical significance was evaluated by one-way ANOVA with LSD's post hoc test (**d**–**f**). Source data are provided as a Source Data file.

impairs the mTORC1-mediated TFEB phosphorylation in a Rag GTPase-dependent manner[61]. We found that 24 h-U exposure at 600 μM, but not at 100 μM, inhibited the mTORC1 activity with the impairment of phosphorylation of S6 kinase (S6K), a canonical mTORC1 substrates, and that 24 h-U exposure at 100 or 600 μM plus ML-SA1 treatment had the same effect on the mTORC1 activity as 24 h-U exposure at 100 or 600 μM alone, whereas ML-SA1 had no effect on the level of phosphorylated S6K protein in control HK-2 cells (Fig. 6k, l), suggesting that ML-SA1 had no effect on mTORC1 activity. U-exposure at high U concentration (600 μM) likely caused the general cellular toxicity, leading to the inhibition of mTORC1-mediated phosphorylation of TFEB and S6K. As mTORC1-mediated phosphorylation of TFEB is dependent on RagC/D GTPases[81,82], we expressed an HA-tagged constitutively active mutant of RagC (S75L) in HK-2 cells to test the effect of RagC on U-induced TFEB nuclear translocation. Overexpression of constitutively active RagC GTPases significantly inhibited TFEB nuclear translocation induced by 24 h-U exposure at 100 and 600 μM and this inhibition was again reverted by concomitant treatment with mTORC1 inhibitor Torin1[89] (Supplementary Fig. 9e, f), indicating that 24 h-U exposure alone impairs the mTORC1-mediated phosphorylation of TFEB by inactivating RagC GTPase, which is associated with the specific effect of lysosomal damage and subsequent TRPML1 activation on RagC/D activity[61]. While RagC activity is dispensable for mTORC1-mediated S6K phosphorylation[61,81,82,90,91], RagA/B GTPases are major modulators of S6K phosphorylation through recruiting mTORC1 to the lysosomal surface[92,93]. Therefore, the inhibitory effect on S6K caused by U exposure at the high U concentration is likely due to toxicity-induced inhibition of RagA/B. Moreover, U-induced inactivation of RagC and impairment of the recruitment of mTORC1 and TFEB to lysosomes mediate the inhibition of mTORC1-mediated phosphorylation of TFEB, leading to the TFEB nuclear translocation and activation and subsequent transcriptional activation of TRPML1. Importantly, TFEB overexpression/activation promotes lysosomal exocytosis via the transcriptional activation of TRPML1[50]. We propose that the effects of U-induced TFEB nuclear translocation and subsequent TRPML1-mediated lysosomal exocytosis are an ego defense mechanism against U cytotoxicity. Nakamura, et al reported a beneficial role of TFEB activation for preventing oxalate-mediated nephropathy which accompanies lysosomal damage in the proximal tubule of the kidneys, probably through TRMPL1-mediated $Ca^{2+}$ efflux from lysosomes[61]. However, the effect of self-defense mechanism is not sufficient to resist U-toxicity as the U accumulation period is positively associated with the U-toxicity[3]. On the other hand, sustained abnormal TFEB activation is known to be a relatively common cause of Birt–Hogg–Dubé (BHD)-associated renal tumors and renal cell carcinoma[81,82,94]. Therefore, administration of the TRPML1 agonist promotes the excretion of U from cells, which in turn reduces U-induced cytotoxicity and long-term effects of U on carcinogenesis.

To confirm the dual roles of ML-SA1 in promoting the removal of intracellular U and reducing the U-induced cell death through the activation of the TRPML1-TFEB pathway, the effects of TRPML1 and TFEB knockdown were evaluated. We used specific shRNAs to knock down the expression of TRPML1 and TFEB in HK-2 cells (Supplementary Fig. 8a, b, Supplementary Fig. 10a, b). Both TRPML1 knockdown and TFEB knockdown significantly reduced the level of β-hex in the culture medium of U-loaded HK-2 cells after 24 h-U exposure at 50, 100, and 600 μM, and completely abolished the ML-SA1-induced β-hex release from U-loaded HK-2 cells after 24 h-U exposure at 50, 100, and 600 μM (Fig. 7a, c). Consistently, knockdown of TRPML1 or TFEB robustly increased the intracellular U content and significantly decreased the level of extracellular U in the culture medium of the U-loaded HK-2 cells after 24-h U exposure at 50, 100, and 600 μM, and abolished ML-SA1-induced efflux of intracellular U (Fig. 7b, d). Correspondingly, Calcein-AM/PI staining assay revealed that TRPML1 or TFEB knockdown significantly increased U-induced cell death and greatly abolished the effect of ML-SA1 on the reduction of U-induced cell death in response to 24 h-U exposure at 100 and 600 μM in U-loaded HK-2 cells (Fig. 7e, f and Supplementary Fig. 10c, d). These results further demonstrate that the effects of TRPML1 or TFEB knockdown and/or ML-SA1 on intracellular U content were positively associated with their effects on U-induced cell death. The effects of TFEB knockdown on abolishing ML-SA1-activated lysosomal exocytosis attribute to a very low level of TRPML1, resulting from the positive feedback loop between TRPML1 and TFEB in TFEB-depleted cells. We thus propose that bi-effects of ML-SA1 on clearing the intracellular U and subsequently decreasing the U-induced cell death are dependent on both lysosomal exocytosis and biogenesis through forming a positive feedback loop between TRPML1 and TFEB.

It is noted that 24 h-U exposure alone at 100 and 600 μM significantly increased the cell death rate although the lysosomal exocytosis was enhanced, as judged by increased levels of β-hex and U in the culture medium of U-loaded HK-2 cells (Figs. 4c, d and 7e, f). These results support the notion that the self-defense in U-loaded HK-2 cells is not sufficient to resist U-toxicity with a long-term U accumulation[3].

### TRPML1 activation promoted the removal of intracellular U via triggering TRPML1-mediated lysosomal exocytosis in HK-2 cells after long-term U exposure

Long-term exposure of HK-2 cells to U at 1, 5, and 10 μM for 10, 20, and 30 days were used to mimic low-dose chronic exposures. Excitingly, 10, 20, and 30 day-delayed ML-SA1 treatment produced a marked reduction of intracellular U in U-loaded HK-2 cells with long-term U-exposures at 1, 5, and 10 μM for 10, 20, and 30 days (Fig. 8). Conversely, TRPML1 knockdown robustly increased the intracellular U content and abolished the ML-SA1-induced removal of intracellular U (Fig. 8). The effect of ML-SA1 on reducing intracellular U content in TRPML1-knockdown HK-2 cells with long-term U-exposures may be associated with the induction of TRPML1 expression in HK-2 cells after long-term U-exposures. These data demonstrate that ML-SA1 promotes the removal of intracellular U via triggering TRPML1-mediated lysosomal exocytosis after chronic U exposure.

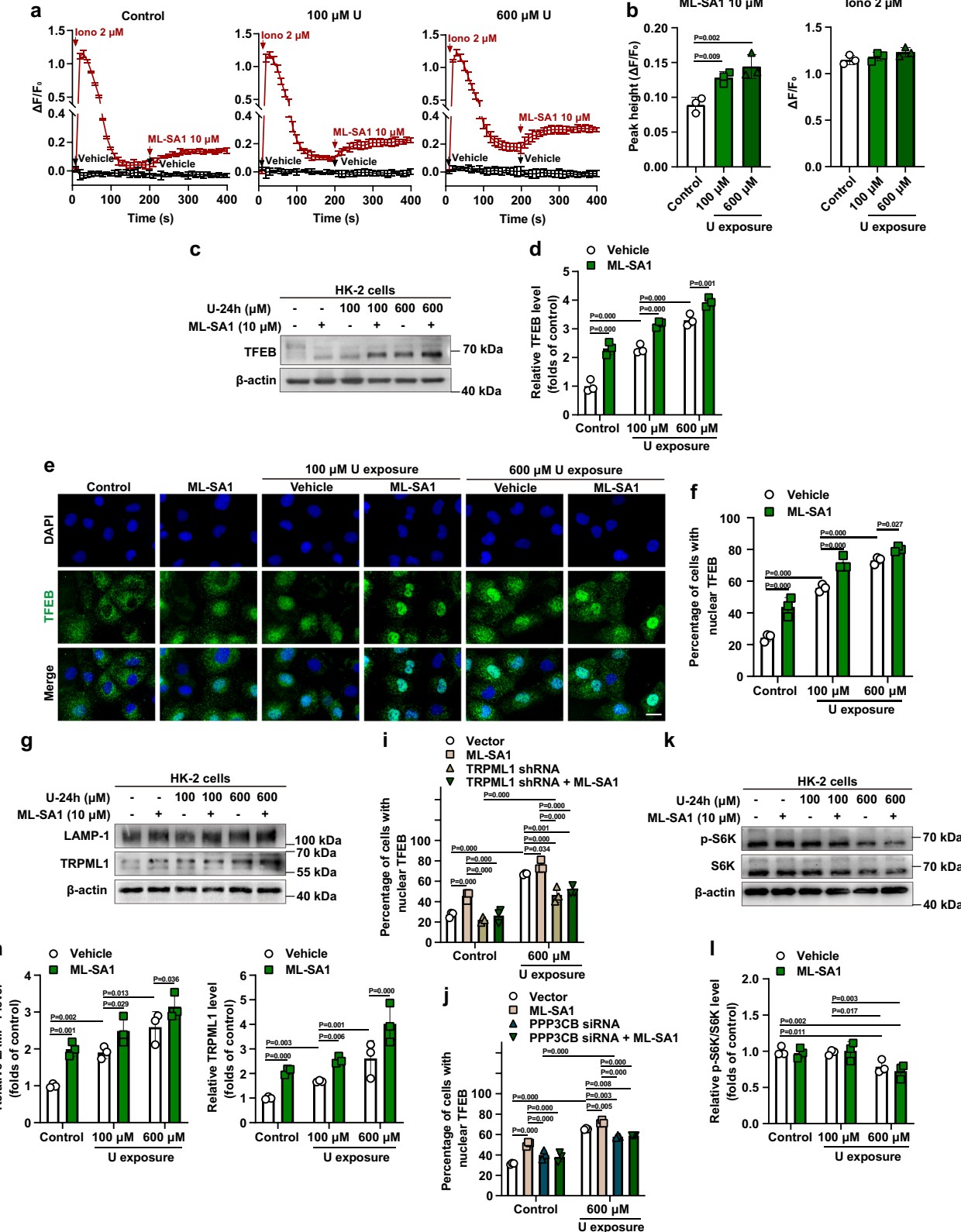

## TRPML1 activation is associated with the increase of autophagic clearance of U-containing damaged organelles and macromolecular complexes in U-loaded HK-2 cells

Using transmission electron microscopy (TEM) analyses, we further studied the status of U in HK-2 cells after short-term or long-term U exposure. Studies have revealed that U is present in a variety of the cultured cells including PTECs as either a soluble form located in the cytoplasm and nucleus or a precipitate form mainly in the lysosomes, depending on U exposure concentration and time[15,16,18–24]. Our TEM analyses showed that U precipitates with needle-like structures were found in autolysosomes and multivesicular bodies and were grown in a U concentration-dependent manner in HK-2 cells after 24 h exposure to U at 50, 100, 300, and 600 μM (Supplementary Fig. 11a). Lysosomes are known to receive inputs through the autophagic and endocytic

**Fig. 6 | TRPML1 activation with ML-SA1 promotes lysosomal TRPML1-mediated $Ca^{2+}$ release and nuclear translocation of TFEB and consequent lysosomal biogenesis in U-loaded renal epithelial HK-2 cells.** For experiments **a**, **b**, HK-2 cells were treated with U at 0, 100, and 600 μM for 24 h. After washout of U, cytosolic $Ca^{2+}$ was measured after treatment with ionomycin (2 μM) followed ML-SA1 (10 μM) by Fluo-4 imaging. For experiments **c–l**, HK-2 cells or HK-2 cells transfected with either TRPML1 shRNA or PPP3CB siRNA or corresponding empty vector plasmid or scramble control siRNA were exposed to U at 0, 100 and/or 600 μM for 24 h. After washout of U, the cells were treated with vehicle or ML-SA1 (10 μM) for 30 min and then analyzed. **a** ML-SA1–induced lysosomal $Ca^{2+}$ release in control and U-loaded HK-2 cells treated with 100 and 600 μM U for 24 h. **b** Quantification of cytosolic $Ca^{2+}$ peak values shown in **a**. **c**, **d** Western blotting analysis of TFEB in HK-2 cells after U exposure and ML-SA1 treatment.

**e** Representative images of immunofluorescence staining of TFEB (green) in HK-2 cells after U exposure and ML-SA1 treatment. Images share the same scale bar (20 μm). **f** Quantitative analysis of nuclear TFEB under various treatment conditions shown in **e**. **g**, **h** Western blotting analysis of LAMP-1 and TRPML1 in HK-2 cells after U exposure and ML-SA1 treatment. **i**, **j** Quantitative analysis of nuclear TFEB in TRPML1- or PPP3CB-knockdown HK-2 cells after U exposure and ML-SA1 treatment. Representative images of immunofluorescence staining of TFEB (green) are shown in Supplementary Fig. 8e, f. **k**, **l** Western blotting analysis of p-S6K and S6K in HK-2 cells after U exposure and ML-SA1 treatment. Iono: ionomycin. **f**, **i**, **j** More than 200 cells were analyzed in each sample. Data represent mean ± SD. $n = 3$ independent experiments. Statistical significance was evaluated by one-way ANOVA with LSD's post hoc test (**b**, **d**, **f**, **h–j**, **l**). Source data are provided as a Source Data file.

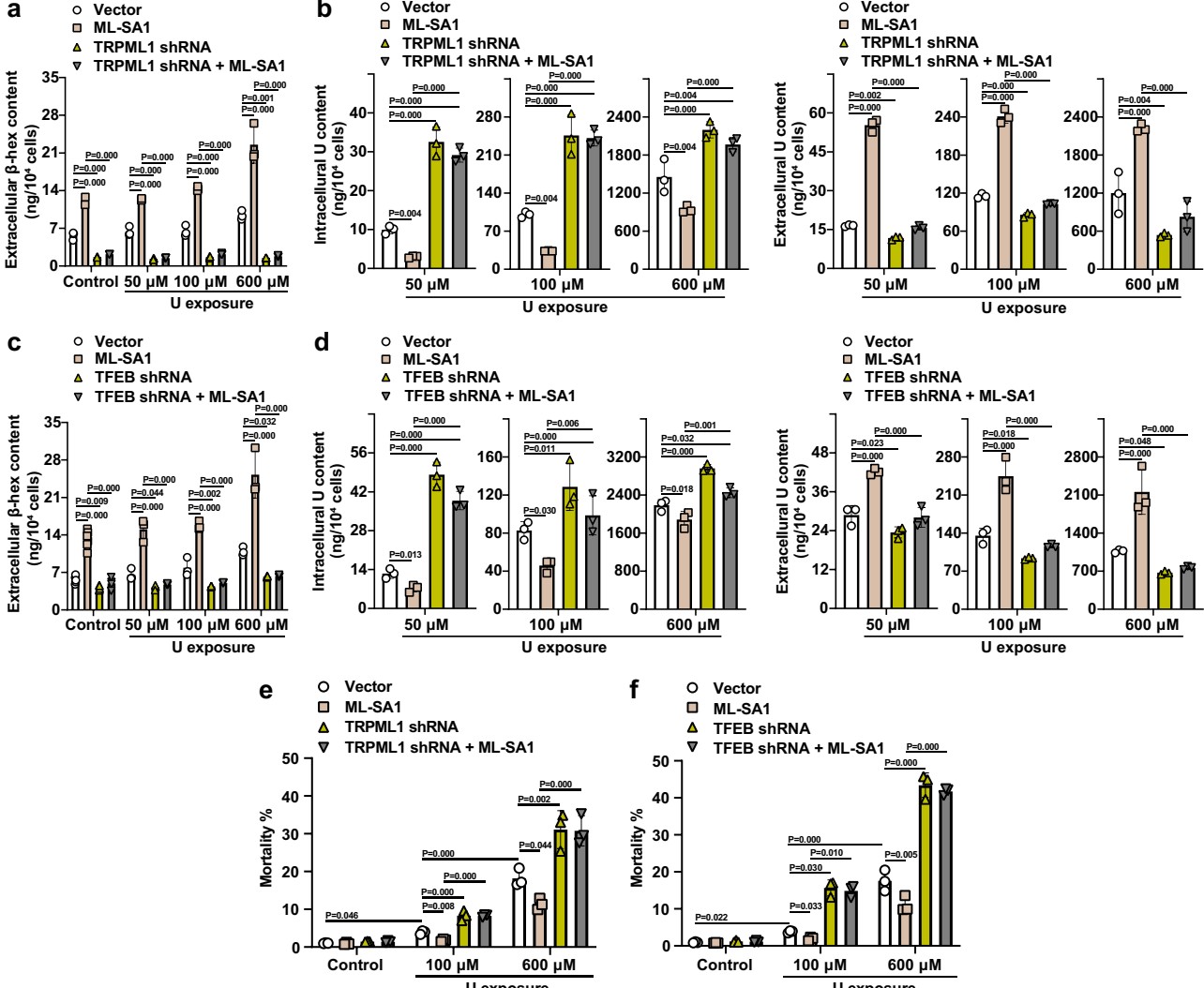

**Fig. 7 | Knockdown of TRPML1 and TFEB abolishes the ML-SA1 effects on the removal of intracellular U and reduction of the U-induced cell death in U-loaded renal epithelial HK-2 cells.** HK-2 cells transfected with either TRPML1 shRNA or TFEB shRNA or corresponding empty vector were exposed to U at 0, 50, 100, and 600 μM for 24 h. After washout of U, the cells were treated with either vehicle or ML-SA1 at 10 μM for 30 min and then analyzed. **a** Comparison of β-hex release from TRPML1-knockdown HK-2 cells or empty vector-transfected HK-2 cells after U exposure and ML-SA1 treatment. **b** Comparison of intracellular and extracellular U contents in TRPML1-knockdown HK-2 cells or empty vector-

transfected HK-2 cells after U exposure and ML-SA1 treatment. **c** Comparison of β-hex release in TFEB-knockdown HK-2 cells or empty vector-transfected HK-2 cells after U exposure and ML-SA1 treatment. **d** Comparison of intracellular and extracellular U contents in TFEB-knockdown HK-2 cells or empty vector-transfected HK-2 cells after U exposure and ML-SA1 treatment. **e**, **f** TRPML1 or TFEB depletion enhances the cell death after U exposure, which was not reversed by ML-SA1 treatment. Data represent mean ± SD. $n = 3$ independent experiments. Statistical significance was evaluated by one-way ANOVA with LSD's post hoc test (**a–f**). Source data are provided as a Source Data file.

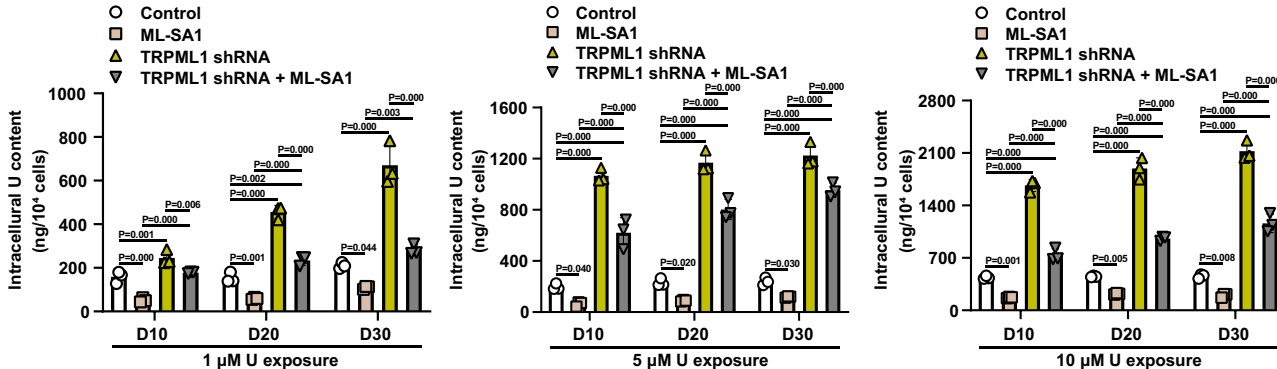

**Fig. 8 | TRPML1 activation with ML-SA1 promotes the removal of intracellular U in U-loaded renal epithelial HK-2 cells after long-term U exposure.** Comparison of intracellular U content in TRPML1-knockdown HK-2 cells or empty vector-transfected HK-2 cells after long-term U exposure (1, 5, 10 μM for 10, 20, 30 days) and the subsequent treatment of ML-SA1 at 10 μM for 30 min after washout of U. Data represent mean ± SD. $n = 3$ independent experiments. Statistical significance was evaluated by one-way ANOVA with LSD's post hoc test. Source data are provided as a Source Data file.

pathways[30]. Autolysosomes are from the fusion of autophagosomes with lysosomes or endosomes to accomplish autophagic degradation of unwanted cellular components[95,96]. Multivesicular bodies are a special kind of late endosomes coming from early endosomes in a series of maturation processes of endocytic vesicles in the endo-lysosomal system. Multivesicular bodies fuse with lysosomes to become endolysosome hybrids to mediate the degradation of extra-cellular material internalized by endocytosis/phagocytosis[96]. Our results suggest that U in the lysosomes might come from both autophagy for U-bound dysfunctional cellular components and endocytosis for extracellular U. However, no intracellular U pre-cipitates were seen by TEM in HK-2 cells with U exposure at 10 μM for 10, 20, and 30 d (Supplementary Fig. 11b), which might be due to too low U concentration for forming the U precipitates as reported in human dopaminergic cells[97]. Unexpectedly, ML-SA1 treatment sig-nificantly decreased the intracellular U content via TRPML1-mediated lysosomal exocytosis in U-loaded HK-2 cells after exposure to U at 1, 5, and 10 μM for 10, 20, and 30 d (Fig. 8), indicating that soluble U is also present in the lysosomes. We thus propose that TRPML1 activation by ML-SA1 has another important role in enhancing U enrichment in the lysosomes and protecting against U-induced cytotoxicity by increasing the autophagic clearance of U-containing damaged organelles and macromolecular complexes before promoting the lysosomal exocy-tosis. More details await further studies in the future.

In summary, our data demonstrate that pharmacological activa-tion of TRPML1 by delayed treatment of a single dose of ML-SA1, a potent specific TRPML1 agonist, markedly enhances the removal of U accumulated in the kidney via urinary U excretion, and mitigates the U-induced renal proximal tubular injury and LMP in renal PTECs of mice after single and repeated exposure to low or high dose of U. Consistent with our initial hypothesis, TRPML1 activation by ML-SA1 stimulates the removal of the intracellular accumulated U and reduces the U-induced LMP and cell death, which is mediated by increased lysosomal exocytosis triggered by increased lysosomal Ca$^{2+}$ release. Meanwhile, TRPML1 activation by ML-SA1 promotes the TFEB nuclear translocation via the activation of calcineurin triggered by lysosomal Ca$^{2+}$ release via TRPML1, leading to compensatory lysosomal biogen-esis to benefit the lysosome homeostasis, and in turn activate the TRPML1-mediated lysosomal exocytosis (Fig. 9).

The current study provides the first evidence that TRPML1 activation-induced lysosomal exocytosis and biogenesis are a rational and feasible approach in promoting the clearance of intracellular accumulated U and subsequent reducing U-induced LMP and cell death in renal PTECs in vivo and in vitro. The action of TRPML1-agonism is dependent on the formation of the positive TRPML1 and TFEB feedback loop. The advantage of TRPML1 approach is high efficacy particularly in the delayed treatment after acute or chronic U exposure. TRPML1 is an attractive druggable target for U decorporation and detoxification for acute and chronic U exposure, which can also be extended to decorporation and detoxification of other toxic heavy-metals concentrated in the lysosomes of renal PTECs.

## Methods
### Animals
The 7-week-old male BALB/c mice (20 g ± 2 g) were obtained from Shanghai Jiesijie Laboratory Animal Technology Co. Ltd. (China). The mice under study were maintained in a specific pathogen free (SPF), environmentally suitable barrier system at 20–26 °C and 40–70% humidity with a light/dark cycle of 12/12 h at the Laboratory Animal Center for Drug Evaluation of School of Pharmacy, Fudan University. Mice were fed with a gamma-irradiated AIN-93G purified diet for-mulated for gestating and growing rodents (Wuxi Fanbo Biotechnol-ogy Co. Ltd., China, #FB-D10012G), and water ad libitum. All animal experiments were performed according to the guidelines and proce-dures approved by the Animal Research Ethics Committee of the School of Pharmacy of Fudan University (No. 2021-01-FYS-CHH-013). All mice used in this study were males as it is not expected that sex and gender may affect the effect of TRPML1 agonist ML-SA1 on U-induced nephrotoxicity.

### U exposure and ML-SA1 treatment in mice
The U stock solution was prepared in aqueous uranyl acetate−NaHCO$_3$ solution with uranyl acetate to NaHCO$_3$ molar ratio = 1:10, pH-7.0, as described previously[24]. Mice were injected intramuscularly with a sin-gle dose of U at 0.4 mg/kg or 2.0 mg/kg body weight (bw) or multiple doses of U at 80 μg/kg once daily for 5 days. The mice with single- or multiple-dose of U administration were then randomly divided into three groups. Group 1–2: the mice were treated with intraperitoneal injection of ML-SA1 (Sigma-Aldrich/Merck, USA, #SML0627) (dissolved in deionized water containing 0.88‰ DMSO) at 400 μg/kg or 800 μg/kg bw at 24 h after im injection of U. Group 3: the mice were treated with vehicle (deionized water containing 0.88‰ DMSO). In addition, the mice in blank control group without U exposure were treated with vehicle. Mice were housed in individual plastic metabolism cages to collect urine every 24 h. After the ML-SA1 treatment for 24 h, the mice were anesthetized with an intraperitoneal injection of 2,2,2-tri-bromoethanol at 250 mg/kg (Sigma-Aldrich/Merck, #T48402). Blood samples were collected from the orbital venous sinus in the mice and the serum was separated by centrifuged at 3000 rpm for 5 min. The kidneys from both sides were obtained after perfusion with PBS at 4 °C. Urine and one kidney from each mouse were for U content detection,

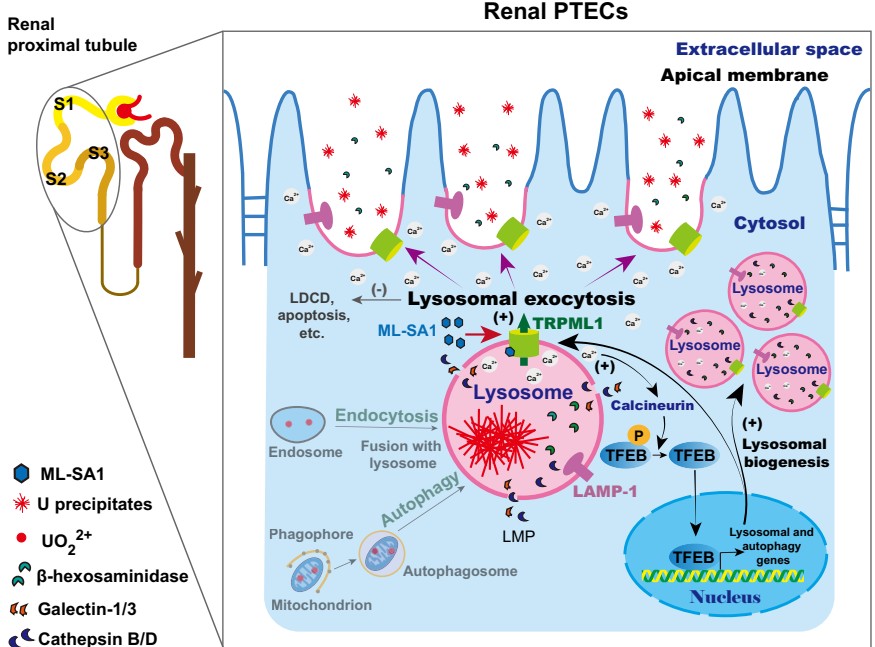

**Fig. 9 | A schematic model for the mechanism by which TRPML1 activation with ML-SA1 promotes the intracellular U clearance and reduces the U-induced LMP and cell death in U-loaded renal PTECs after acute and chronic U exposure.** TRPML1 agonist ML-SA1 activates TRPML1 channels on the perimeter membranes of lysosomes, inducing lysosomal $Ca^{2+}$ release and lysosomal exocytosis. Meanwhile, $Ca^{2+}$-bound calcineurin dephosphorylates TFEB[49], which then translocates to the nucleus to activate the transcription of CLEAR genes to enhance the lysosomal biogenesis and function[43,44]. In turn, TFEB activation may also further promotes the TRPML1-mediated release of lysosomal $Ca^{2+}$ and induction of lysosomal exocytosis[50]. Subsequently, lysosomal exocytosis is promoted, which facilitates the removal of intracellular U and clearance of damaged lysosomes with LMP under the cooperation of compensatory lysosomal biogenesis and consequently reduces the U-induced cell death.

and the other kidney was fixed with 4% paraformaldehyde for paraffin embedding section.

### Cell culture, U exposure, and drug treatments

Human renal proximal tubular epithelial cell line HK2 (#SCSP-511) was purchased from the Type Culture Collection of the Chinese Academy of Sciences, Shanghai, China. The cells were cultured in Dulbecco's Modified Essential Medium and Ham's F-12 Medium (DMEM/F-12) (Gibco, USA, #11320082) containing 10% fetal bovine serum (FBS) and 100 U/ml of penicillin and 100 µg/ml of streptomycin (Gibco, #15140-122) in 5% $CO_2$ at 37 °C. HK-2 cells were exposed to U and treated with ML-SA1 and Vacuolin-1 (Selleck, USA, #S6912) under various conditions as described in the figure legends.

### Knockdown of TRPML1, PPP3CB and TFEB in HK-2 cells

TRPML1 (MCOLN1) shRNA (#TR303307) and empty vector (#TR30012) plasmids were from OriGene (USA). Human PPP3CB siRNA (#s11004) and scramble (SCRMBL) siRNA oligonucleotides (#4390843) were from Thermo Fisher Scientific (USA). TFEB shRNA (#P31675) and empty vector (#P0684) plasmids were purchased from Miaolingbio (China). HK-2 cells were seeded into 12-well plates ($1 \times 10^5$ cells/well). After culture for 14–18 h, the cells were transfected with TRPML1 (MCOLN1) shRNA, TFEB shRNA or corresponding empty vector, PPP3CB siRNA oligonucleotides or scramble siRNA oligonucleotides using Lipofectamine 3000 transfection reagent (Thermo Fisher Scientific, #L3000015) according to the manufacturer's instructions. Western blotting analyses confirmed the efficacy of shRNA- or siRNA-mediated knockdown of each protein.

### U detection in mouse samples

U levels in urine and kidney tissues were determined after wet acid digestion. The digestion solution was a mixture of nitric acid and perchloric acid at a ratio of 3:1 (v/v). Kidney tissue (~0.2 g) was digested with 3 ml of digestion solution at 120 °C for 2 h, and then diluted to 25 mL with deionized water. Each urine sample was directly mixed with 3 ml of digestion solution and then diluted to 25 mL with deionized water. All the samples were further diluted 10–50 times with 2% nitric acid (v/v), and U concentrations were determined by an inductively coupled plasma-mass spectrometry (ICP-MS) with Nexlon 1.5 software (NexION300X, PekinElmer, USA). Qualitative data were presented as micrograms per g tissue.

In addition, U content in the kidney of mice with/without perfusion was examined by ICP-MS. It was found that the perfusion at 48 h after intramuscular injection of single-dose U and at 48 h after the last intramuscular injection of multiple-doses U had no effects on U content in kidney of U-exposed mice (Supplementary Table 1).

### Renal function test

The levels of serum creatinine (CRE) and blood urea nitrogen (BUN) were measured by commercial kits (FUJIFILM Wako Pure Chemical Corporation, Japan, #995-32591, #998-17701) following the manufacturer's instructions using an automatic biochemistry analyzer (7180, Hitachi, Japan).

### Immunohistochemical (IHC) analysis

Mouse kidney tissues were fixed in 4% paraformaldehyde for 24 h, embedded in paraffin, and cut into 4 µm thick sections. The kidney tissue sections were deparaffinized, rehydrated, blocked, and then incubated with the following specific primary antibodies: anti-KIM-1 (1:400, LifeSpan BioSciences, USA, #LS-B2103), anti-LAMP-1 (1:200, Invitrogen, USA, #14-1071-82), anti-TRPML1 (1:200, Atlas Antibodies, USA, #HPA031763) and anti-Galectin-1 (1:200, HuaBio, China, #ET1705-83). After overnight incubation at 4 °C, the sections were incubated with horseradish peroxidase (HRP)-polymer-conjugated secondary antibody (No dilution, Proteintech, USA, #PK10006) at room temperature for 1 h and stained with 3,3-diaminobenzidine solution and

hematoxylin. At least 20 random fields of each section were examined and photographed through a digital microscope with ZEN Blue Imaging Analysis 2016.08.06 software (Imager M2, ZEISS, Germany). The number of KIM-1-positive proximal tubules or number of LAMP-1 on the apical membrane-positive proximal tubules, the intensity of TRPML1 positively stained area on the apical membrane of the proximal tubules and the intensity of galectin-1-positive proximal tubules per high power field were quantitatively analyzed with ImageJ 1.8.0 software (National Institute of Health, USA).

## Pathological analysis

The paraformaldehyde-fixed paraffin-embedded mouse kidney tissue sections were stained with hematoxylin and eosin (H&E). At least 20 random fields of each section were examined and photographed with a digital microscope (Imager M2, Zeiss). The number of injured renal proximal tubules with necrotic or exfoliated cells in each high power field of view were determined.

## TUNEL assay

The levels of apoptosis were detected by the TUNEL FITC apoptosis detection kit (Vazyme Biotech Co. Ltd., China, #A111-03) in the paraformaldehyde-fixed paraffin-embedded mouse kidney tissue sections. At least 20 random fields of each section were examined and photographed with a laser scanning confocal microscope (Leica Application Suite X 3.5.6.21594 software) (SP8, Leica, Germany). The number of green fluorescent-labeled apoptotic cells in each high-power field was counted, or the green fluorescence intensity in each high power field was quantitatively analyzed with ImageJ 1.8.0 software.

## Measurement of β-hexosaminidase (β-hex) activity in HK-2 cells

After the U exposure and drug treatments of HK-2 cells, the culture media were collected, centrifuged at 1000 rpm for 5 min to remove dead cells and debris and diluted 20 times with sodium citrate-phosphate buffer (pH 4.5). The diluted culture supernatant (350 μl) was incubated with 50 μl of 6 mM 4-methyl-umbelliferyl-N-acetyl-β-D-glucosaminide (Sigma-Aldrich/Merk, #M2133) in sodium citrate-phosphate buffer (pH 4.5) for 60 min. After stopping the reaction with 100 μl of 2 M $Na_2CO_3$-1.1 M glycine buffer, the fluorescence was measured with excitation at 365 nm and emission at 450 nm by a multifunctional microplate reader (Synergy H1, Biotek, USA). β-hex content was calculated by using recombinant β-hex calibration curves (MyBioSource, USA, #MBS169556).

## Evaluation of intracellular and extracellular U contents of HK-2 cell

After U exposure and drug treatments of HK-2 cells, the culture media was collected and mixed with alkaline lysis solution in a ratio of 1 to 1 (v/v). The cells were rinsed with PBS containing 10 mM sodium bicarbonate, trypsinized, counted, and centrifuged at 1000 rpm for 5 min. The cell pellets were lysed with alkaline lysis solution at 37 °C for 1 h, as described previously[98]. The U concentrations in culture media and cell lysates were determined by ICP-MS (Nexion300X, PekinElmer). Qualitative data were presented as ng per $10^4$ cells.

## Immunofluorescence staining

For cell surface staining of LAMP-1, HK-2 cells grown on chamber slides were incubated with anti-rabbit LAMP-1 (1:200, Cell Signaling Technology, USA, #9091) for overnight at 4 °C. After rinsed in PBS, cells were further incubated with Alexa Fluor 555-conjugated donkey anti-rabbit secondary antibody (1:500, Invitrogen, USA, #A31572) at room temperature in the dark for 1 h. For intracellular proteins immunofluorescence staining, HK-2 cells cultured on the chamber slides or 24-well plates were fixed with 4% paraformaldehyde for 15 min, permeabilized with 0.5% (v/v) Triton X-100 in PBS for 10 min, and blocked with

10% (v/v) FBS in PBS for 60 min at 37 °C. The cells were then incubated overnight at 4 °C with the following primary antibodies: anti-TRPML1 (1:200, Atlas Antibodies, #HPA031763), anti-Galectin-3 (1:200, BD Biosciences, USA, #556904), anti-LAMP-1 (1:200, Cell Signaling Technology, #9091), anti-cleaved caspase-3 (1:400, Cell Signaling Technology, #9664) and anti-TFEB (1:200, Cell Signaling Technology, #4240). The cells were then stained with appropriate secondary antibodies including Alexa Fluor 555-conjugated donkey anti-rabbit (1:500, Invitrogen, #A31572), Alexa Fluor 555-conjugated donkey anti-mouse (1:500, Invitrogen, #A31570), Alexa Fluor 488-conjugated donkey anti-rabbit secondary antibody (1:500, Invitrogen, #A21206) or Alexa Fluor 488-conjugated donkey anti-mouse secondary antibody (1:500, Invitrogen, #A21202) at room temperature in the dark for 1 h. After the secondary antibody incubation, nuclei were counterstained with DAPI-containing mounting media (Santa Cruz Biotechnology, USA, #sc-24941). The images were captured with a fluorescence microscope (Imager M2, ZEISS), a laser scanning confocal microscope (SP8, Leica), or an ImageXpress Micro 4 screening system (MetaXpress 6.6.1.42 software) (Molecular Devices, USA) and analyzed with ImageJ 1.8.0 software, and the number of cells analyzed per sample in each experiment was indicated in the figure legends. Qualitative data were presented as Manders' colocalization coefficient or percentages or folds of the corresponding control.

## Calcein-AM/PI staining assay

HK-2 cells were seeded into 24-well plates with $5 \times 10^4$ cells per well. After culture overnight, the cells were treated with different reagents as described in the figure legends. The cells were then stained with calcein-AM and pyridine iodide (PI) (Dojindo, Japan, #C542) for 15 min at 37 °C, and examined with an inverted fluorescence microscope (ECLIPSE Ts2-FL, Nikon, Japan) operated by CapStudio software (iMG, version biology 3.7.2). About 1000 cells were analyzed in each sample and the percentages of cell death were calculated.

## Fluo-4 Ca²⁺ imaging

HK-2 cells were seeded into 384-well microplates with $8 \times 10^3$ cells per well and grown overnight. After exposure to U at 0, 100, and 600 μM for 24 h, the cells were incubated with 25 μL of the dye loading solution containing Fluo-4 AM and probenecid at 37 °C for 30 min, then at room temperature for an additional 30 min using Fluo-4 NW Calcium Assay Kit (Thermo Fisher Scientific, #F36206) according with the manufacturer instruction. Fluorescence intensity was recorded at excitation wavelength of 470 nm and emission wavelength of 516 nm by an FLIPR Penta high-throughput real-time fluorescence imaging analysis system (FLIPR) operated by FLIPR Penta (Molecular Devices). The changes of Fluo-4 fluorescence ΔF over basal fluorescence $F_0$ ($\Delta F/F_0$) were used to monitor the changes of cytosolic [Ca²⁺] upon stimulation. Ionomycin (2 μM, Yeasen Biotechnology Shanghai, China, #50401ES03) was added at the beginning of all experiments to induce Ca²⁺ release from ER Ca²⁺ stores[77,78]. ML-SA1 at 10 μM was added after ionomycin treatment to induce the Ca²⁺ release from lysosomes.

## Western blotting analyses

Cell lysates were prepared with ice-cold radioimmunoprecipitation assay (RIPA) buffer with 1 mM phenylmethylsulfonyl fluoride (PMSF), and 30 μg of protein was subjected to SDS-polyacrylamide gel electrophoresis (SDS-PAGE) followed by electro-transfer onto polyvinylidene difluoride (PVDF) membrane (Millipore, USA, #ISEQ00010). The membranes were blocked for 1 h with 5% skim milk in Tris-buffered saline/Tween (TBST), and incubated with a primary antibody overnight at 4 °C followed by a HRP-conjugated secondary antibody (1:1000, Beyotime Biotechnology, China, #A0208) or a HRP-conjugated secondary antibody (1:1000, HuaBio, #HA1006) for 1 h. The bands were visualized with an ECL detection kit (Millipore, #WBKLS0100) and analyzed by the BIO-RAD ChemiDoc XRS system (BIO-RAD, USA). The

primary antibodies used in this study are anti-LAMP-1 (1:1000, Cell Signaling Technology, #9091), anti-TRPML1 (1:1000, Atlas Antibodies, #HPA031763), anti-phospho-p70 S6 kinase (1:1000, Cell signaling Technology, #9234), anti-p70 S6 kinase (1:1000, Cell signaling Technology, #9202), anti-TFEB (1:1000, Cell Signaling Technology, #4240), anti-β-actin (1:1000, Cell signaling Technology, #4970) and anti-Vinculin (1:1000, Cell signaling Technology, #4650).

## Statistical analysis

All data were expressed as average ± standard deviation (SD) of independent experiments or samples. The data were analyzed using SPSS 20.0 software (IBM SPSS, Somers, NY, USA). Comparison between the two groups was completed using Student's two-tailed $t$-test. Multiple comparisons were carried out using one-way ANOVA with LSD's post hoc test. A $P$ value of less than 0.05 was considered to indicate statistical significance.

## Reporting summary

Further information on research design is available in the Nature Portfolio Reporting Summary linked to this article.

## Data availability

All data supporting the findings of this work are available within the paper and its Supplementary information files. Source data are provided in this paper.

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

## Acknowledgements

We thank Qiang Liu (School of Public Health/Key Laboratory of Public Health Safety of Ministry of Education/Collaborative Innovation Center of Social Risks Governance in Health, Fudan University) and Yulan Li (Shanghai Institute of Applied Physics, Chinese Academy of Sciences) for technical assistance in ICP-MS detection. This work was supported by a grant from the National Natural Science Foundation of China (Grant No. 81972971).

## Author contributions

H.C. developed the research concept and designed the experiments. D.Z., R.W., and H.Z. conducted the experiments and data analyses. M.W. contributed to the literature research. X.Z. ordered experimental materials and supplies. H.C. wrote the manuscript and all authors read and approved the final manuscript. H.C. acquired funding.

## Competing interests

The authors declare no competing interests.
