## [Peer Review File · Nature Communications]

Induction of lysosomal exocytosis and biogenesis via TRPML1 activation for the treatment of uranium-induced nephrotoxicityREVIEWER COMMENTS

Reviewer #1 (Remarks to the Author):

1. The quality of immunohistochemical staining is not impressive. For example, in figure 1C, the blue (nucleus) color looks everywhere, which makes people hard to see the change in KIM-1 level. Same problem occurred in other IHC figures.
2. In figure 1C, the rectangle of im 2.0 mg/kg U-vehicle group is incorrect.
3. ML-SA5 is a novel TRPML1 channel agonist, which is more potent and water-soluble than ML-SA1. I am wondering why the authors did not use ML-SA5 instead of ML-SA1.
4. In animal studies, in addition to TRPML1 channel agonist, TRPML1 channel inhibitor such as ML-SI1 can be used to test the hypothesis in a different direction.
5. In figure 5A, it is better to take high-resolution images by confocal microscopy or super-resolution microscopy to determine the colocalization of galectin-1 and Lamp-1. Since vacuolin-1 can remarkably enlarge lysosomes, the low-magnification images are meaningless.
6. It is interesting that U and ML-SA1 both can increase the expression of Lamp-1 and TRPML1 in HK-2 cells. However, the expression is not equal to the function. It is extremely important for the authors to determine whether U had impact on TRPML1 channel-mediated Ca²⁺ release. GCaMP3 Ca²⁺ imaging and whole-lysosome patch clamping are both good methods for this goal. If the authors cannot perform these 2 experiments, please read the paper entitled "Lysosomal Ca²⁺ Homeostasis and Signaling in Health and Disease" to learn how to record TRPML1 channel-mediated Ca²⁺ release using Fura-2. Measurement of intracellular Ca²⁺ concentration by Fura-2 after chronic stimulation is meaningless. Simultaneous recording of TRPML1 channel activity in response to U and ML-SA1 is necessary.

Reviewer #2 (Remarks to the Author):

The paper from Zhong D et al. investigates the efficacy of a new decorporating agent for uranium intoxication of mice or kidney cell model. The work is very well designed, and the results obtained bring new additional knowledges on the uranium renal effects at the tissue and cell level in comparison to the litterature.

General comments:

The topic is of high interest, especially at low doses where the knowledge on the cellular and molecular effects need to be improved. The authors have well compared their results to the literature and bring new and interesting knowledge on uranium toxicological mechanisms and treatment. With some clarification and additional information's in the result and discussion sections, the paper could be improve before publication.

As a general comment, the effect of ML-SA1 compared to other existing decorporating agents cited by the authors should be experimentally test or discussed. The hypothesis is based on the cellular distribution of uranium in lysosomes but as shown by others it is also localized in nuclear or cytoplasmic cell compartments due to other mechanisms of entry and binding to different proteins or DNA; it should be further discussed.

Specific comments:

Abstract

The first sentence should be modified as U forms precipitates only in some conditions, especially at high (cytotoxic) concentration.

Introduction:

Page 3, Line 50-54: Uranium is also known to localize in cells and tissue as non-precipitated (soluble) forms, it should be cited in the introduction.

Page 5, Line 94-95: The sentence "Our studies indicate..." isn't related to any citation hereafter.

Results and Discussion:

Page 6, line 104: The term "chronic exposure" should be replaced by "repeated exposure"

Page 6, line 108: The term "low dose" for 0.4 mg/kg is not really appropriate for this range of dose as it is a nephrotoxic dose

Figure 1: Treatment with ML-SA1 alone is necessary for a better comprehension of the effects on

KIM-1 immunostaining, and histopathological effect.

Page 6, line 113-114: The sensitivity of the markers of function CRE and BUN should be discussed as they are known to be augmented only when high injury of the kidney occurs.

Page 8 and figure 1a: How do you explain that U content of the kidney is still very high and correlated to tubular damage in animals treated with ML-SA1. Justification should be added in the text.

Page 9 and figure 1b: How do you explain the absence of dose-response effect of ML-SA1?

Page 9 and figure 1f,g: What kind of injury is scored on PT; necrosis, inflammation, fibrosis? It should be precise in the figure and in the text.

Page 12, line 254: Comparison to literature references on HK-2 cells should be cited here (Prat O 2005).

Page 18, line 377: Comparison to kidney cells should be cited here (Suhard D, 2018)

Page 20, line 427: What is the influence of perfusion on U tissue content? Perfusion is done with formalin?

Materials and methods:

Page 27, TEM: How do you verify that morphology and uranium distribution is preserved by centrifugation before the addition of fixative chemicals?

Reviewer #3 (Remarks to the Author):

In this manuscript, Zhong et al analyze the role of TRPML1-mediated lysosomal exocytosis in the treatment of uranium-induced nephrotoxicity. They show that administration of the TRPML1 agonist ML-SA1 ameliorates the kidney phenotype of mice exposed to single or multiple doses of uranium. ML-SA1 treatment results in decreased intracellular accumulation and enhanced extracellular release of uranium, both in cellular and animal models. Mechanistically, the authors show that ML-SA1 treatment induces lysosomal exocytosis through activation of TFEB.

This is a well-written manuscript and the experimental flow is logically linked. The findings that TRPML1 activation improves uranium-induced kidney injury may be medically relevant. However, a beneficial role for TRMPL1-mediated TFEB activation has already been shown in oxalate nephropathy, in which, similarly to uranium treatment, kidney injury is observed upon oxalate-mediated induction of lysosomal damage (PMID: 32989250). Furthermore, there are a few mechanistic issues that need to be addressed. For instance, the mechanism by which uranium leads to TFEB activation is unclear and requires further elucidation. Finally, there are a few conceptual and technical points that also need to be clarified.

Specific points:

1. The mechanism by which uranium treatment leads to TFEB nuclear translocation is unclear. TFEB is known to be modulated by mTORC1-mediated phosphorylation, via a substrate-selective mechanism that relies on the activity of RagC/D GTPases (PMID: 32612235, 35358174). Does uranium treatment induce TFEB de-phosphorylation? Does it lead to general inhibition of mTORC1 activity (i.e. impaired phosphorylation of its well-characterized substrates S6K and 4E-BP1)? Does the expression of a constitutively active RagC mutant (S75L) prevent uranium-induced TFEB nuclear translocation?
2. Related to the previous point and as stated above, a beneficial role for TRMPL1-mediated TFEB activation has already been shown in oxalate nephropathy (PMID: 32989250). This reference should be added and discussed by the authors. Interestingly, in this model oxalate-induced lysosomal damage was shown to induce TFEB activation through a TRPML1-mediated mechanism that modulates the activity of RagC/D GTPases. Thus, it would be important to assess if uranium promotes TFEB activity through a similar mechanism.
3. The authors show that uranium induces TFEB nuclear translocation, which then leads to TRPML1-mediated lysosomal exocytosis. They also show that TRPML1 activation by ML-SA1 strongly enhances lysosomal exocytosis in HK2 cells in a TFEB-dependent manner, as TFEB depletion blunts ML-SA1-induced lysosomal exocytosis. However, TRPML1 is the major effector of TFEB-induced lysosomal exocytosis and should be functionally downstream of TFEB. How do the authors explain this discrepancy? Are the expression levels of TRPML1, a direct TFEB transcriptional target, too low

in TFEB-depleted cells for proper activation by ML-SA1? Does TRPML1 overexpression promote lysosomal exocytosis, and does it protect from uranium-induced cell death, both in control and TFEB-depleted HK-2 cells?

4. Related to point 3, is the activation of TFEB, using TRPML1-unrelated drugs (e.g. torin treatment), sufficient to promote lysosomal exocytosis and protection from cell death in uranium-treated HK2 cells?

5. The analysis of lysosomal exocytosis in cellular systems has been only done by assessing the secretion of lysosomal enzymes. Additional assays are required for proper characterization of lysosomal exocytosis. For instance, plasma membrane localization of LAMP1, using antibodies recognizing the luminal region of the protein, should be also determined by FACS and immunofluorescence analysis in non-permeabilized cells subjected to uranium and ML-SA1 treatment.

6. Higher quality images for immunofluorescence analyses in cell lines are needed. For instance, in figure 5a cells are too small and can be barely seen. Furthermore, the fluorescence signal is heavily saturated. Thus, no conclusions can be drawn from co-localization analyses using these images. A similar issue also occurs for TFEB localization shown in Figure 6. Higher resolution and higher magnification images need to be provided, using non-saturated signals. Co-localization analyses should be then performed using the newly acquired images only.

7. The number of cells used for the quantifications of cellular assays need to be provided, together with detailed information on the statistical analyses used.

Response to Reviewers

Comments from Reviewer #1:

Q1. The quality of immunohistochemical staining is not impressive. For example, in figure 1C, the blue (nucleus) color looks everywhere, which makes people hard to see the change in KIM-1 level. Same problem occurred in other IHC figures.

Response: We thank the reviewer for bringing up this issue. We re-examined all the tissue sections of IHC staining by a digital microscope and re-captured high resolution images to improve the quality of all IHC images and re-checked all the IHC data (Fig. 1, 2, 3, Supplementary Fig. 1, 2, 4, 5, 6).

Q2. In figure 1C, the rectangle of im 2.0 mg/kg U-vehicle group is incorrect.

Response: Thank you for pointing this out. It has been corrected.

Q3. ML-SA5 is a novel TRPML1 channel agonist, which is more potent and water-soluble than ML-SA1. I am wondering why the authors did not use ML-SA5 instead of ML-SA1.

Response: We agree that ML-SA5 is a more potent TRPML1 agonists for TRPML1. On the other hand, many studies have demonstrated that ML-SA1 can promote lysosomal exocytosis^{62,63,38,31}. Thus, we selected ML-SA1 to test its therapeutic effects in uranium-induced nephrotoxicity. In the future, other TRPML1 agonists including ML-SA5 will be tested.

Q4. In animal studies, in addition to TRPML1 channel agonist, TRPML1 channel inhibitor such as ML-SI1 can be used to test the hypothesis in a different direction.

Response: We thank the reviewer for bringing up this issue. As suggested, we have performed such studies and found that ML-SI1 was unable to increase the renal U accumulation (data not shown), which could be due to its unknown pharmacokinetic (PK) profile. On the other hand, we also used the TRPML1 shRNA to test the hypothesis in a different direction *in vitro*. As expected, knockdown of TRPML1 not

only robustly increased the intracellular U content and U-induced cell death but also abolished ML-SA1-induced efflux of intracellular U and ML-SA1-mediated reduction of U-induced cell death in renal epithelial HK-2 cells (Fig. 7b, e).

Q5. In figure 5A, it is better to take high-resolution images by confocal microscopy or super-resolution microscopy to determine the colocalization of galectin-1 and Lamp-1. Since vacuolin-1 can remarkably enlarge lysosomes, the low-magnification images are meaningless.

Response: As the reviewer suggested, we repeated the experiments described in Fig. 5a, d. We used the fluorescence microscopy (ImageXpress Micro 4, Molecular Devices, San Jose, CA, USA) to capture the cell fluorescence images of the colocalization of galectin-3 and LAMP-1. The quality of images was significantly improved now, and we found that vacuolin-1 can remarkably enlarge lysosomes (Fig. 5a).

Q6. It is interesting that U and ML-SA1 both can increase the expression of Lamp-1 and TRPML1 in HK-2 cells. However, the expression is not equal to the function. It is extremely important for the authors to determine whether U had impact on TRPML1 channel-mediated Ca^{2+} release. GCaMP3 Ca^{2+} imaging and whole-lysosome patch clamping are both good methods for this goal. If the authors cannot perform these 2 experiments, please read the paper entitled “Lysosomal Ca^{2+} Homeostasis and Signaling in Health and Disease” to learn how to record TRPML1 channel-mediated Ca^{2+} release using Fura-2. Measurement of intracellular Ca^{2+} concentration by Fura-2 after chronic stimulation is meaningless. Simultaneous recording of TRPML1 channel activity in response to U and ML-SA1 is necessary.

Response: We thank the reviewer for bringing up this issue. As the reviewer suggested, we used a FLIPR Penta high-throughput real-time fluorescence imaging analysis system (FLIPR) (Molecular Devices, USA) and Fluo-4 NW Calcium Assay Kit (Invitrogen, Thermo Fisher Scientific, USA, Catalog # F36206) to detect the lysosomal Ca^{2+} release. We found that ML-SA1 treatment resulted in a significant cytosolic Ca^{2+}

increase in both control and U-loaded HK-2 cells with 24 h-exposure to U at 100 and 600 μM (Fig. 6a, b), and that cytosolic Ca^{2+} increase induced by ML-SA1 was significantly higher in U-loaded HK-2 cells than in the control cells (Fig. 6a, b).

We further explored the mechanisms underlying U- and ML-SA1-mediated upregulation of LAMP-1 and TRPML1 in HK-2 cells. It is found that knockdown of TRPML1 or PPP3CB (one of the subunits of calcineurin) not only abolished the ML-SA1-induced nuclear translocation of TFEB in both control and U-loaded HK-2 cells but also partially impaired U-induced TFEB nuclear translocation (Fig. 6i, j, Supplementary Fig. 8e, f), suggesting that effect of ML-SA1 on TFEB activation was dependent on TRPML1-mediated calcineurin activation whereas U-induced TFEB nuclear translocation was partly dependent on TRPML1 and calcineurin. In addition, we found that 24 h-U exposure alone inhibited the mTORC1 activity with the impairment of phosphorylation of S6 kinase (S6K), a canonical mTORC1 substrates, and that 24 h-U exposure plus ML-SA1 treatment had the same inhibitory effect on the mTORC1 activity as 24 h-U exposure alone, whereas ML-SA1 had no effect on the level of phosphorylated S6K protein in control HK-2 cells (Fig. 6k, l), suggesting that ML-SA1 had no effect on mTORC1 activity.

TFEB activity is known to be negatively modulated by mTORC1-mediated phosphorylation via a substrate-selective mechanism that relies on the activity of RagC/D GTPases^{81,82}. To test the effect of RagC on U-induced TFEB nuclear translocation, we expressed in HK-2 cells with HA-tagged constitutively active mutant of RagC(S75L). Overexpression of constitutively active RagC GTPases suppressed U-induced TFEB nuclear translocation (Supplementary Fig. 9e, f), indicating that 24 h-U exposure alone inhibited mTORC1-mediated phosphorylation of TFEB. These data demonstrated that 24 h-U exposure and ML-SA1 increase the TFEB nuclear translocation and the expression of TFEB downstream target LAMP-1 and TRPML1 in HK-2 cells by different mechanisms. While 24 h-U exposure increased the TFEB nuclear translocation by inhibiting the activity of mTORC1 and activating the TRPML1 channels in HK-2 cells, ML-SA1 treatment increased the TFEB nuclear translocation via activating the TRPML1 channels in both control and U-loaded HK-2 cells.

Although we were unable to conduct the experiment for recording TRPML1 channel activity, and we found that 24 h-U exposure at 100 and 600 μM alone did not induce changes in the overall levels of cytosolic Ca^{2+} (Fig. 6a). However, we demonstrated that the Ca^{2+} chelator BAPTA-AM inhibited U-induced TFEB nuclear translocation (Supplementary Fig. 9a, b), suggesting that 24 h-U exposure alone could trigger sustained local calcium signals, leading to Ca^{2+} -dependent lysosomal exocytosis (lines 374-380).

Comments from Reviewer #2 (Remarks to the Author):

The paper from Zhong D et al. investigates the efficacy of a new decorporating agent for uranium intoxication of mice or kidney cell model. The work is very well designed, and the results obtained bring new additional knowledges on the uranium renal effects at the tissue and cell level in comparison to the literature.

General comments:

The topic is of high interest, especially at low doses where the knowledge on the cellular and molecular effects need to be improved. The authors have well compared their results to the literature and bring new and interesting knowledge on uranium toxicological mechanisms and treatment. With some clarification and additional information's in the result and discussion sections, the paper could be improve before publication.

As a general comment, the effect of ML-SA1 compared to other existing decorporating agents cited by the authors should be experimentally test or discussed. The hypothesis is based on the cellular distribution of uranium in lysosomes but as shown by others it is also localized in nuclear or cytoplasmic cell compartments due to other mechanisms of entry and binding to different proteins or DNA; it should be further discussed.

Specific comments:

Abstract

Q1: The first sentence should be modified as U forms precipitates only in some

conditions, especially at high (cytotoxic) concentration.

Response: Thank you for pointing out this issue. As the reviewer suggested, we have modified the first sentence as “Uranium (U) is a well-known nephrotoxicant which forms precipitates in the lysosomes of renal proximal tubular epithelial cells (PTECs) after U-exposure at a cytotoxic dose” (line 20).

Introduction:

Q2: Page 3, Line 50-54: Uranium is also known to localize in cells and tissue as non-precipitated (soluble) forms, it should be cited in the introduction.

Response: As suggested, we have added a sentence and cited the references (line 45, lines 51-52).

Q3: Page 5, Line 94-95: The sentence “Our studies indicate...” isn’t related to any citation hereafter.

Response: Thank you for pointing out this issue. Based on our knowledge, we believe that our study reported for the first time that pharmacological activation of TRPML-1 is a promising therapeutic approach for decorporation and detoxification of U-induced nephrotoxicity after single and repeated U exposure.

Q4: Results and Discussion:

Page 6, line 104: The term “chronic exposure” should be replaced by “repeated exposure”

Response: Thank you for your kind suggestion. To mimic chronic exposure, animal experiments used the U repeated exposure method. We have changed the term “chronic exposure” to “repeated exposure” (line 166, line 216, line 495).

Q5: Page 6, line 108: The term “low dose” for 0.4 mg/kg is not really appropriate for this range of dose as it is a nephrotoxic dose

Response: Thank you for pointing out this issue. We have changed the term “low dose” to “toxic dose” for 0.4 mg/kg and “high dose” to “highly toxic dose” for 2 mg/kg” in

the text.

Q6: Figure 1: Treatment with ML-SA1 alone is necessary for a better comprehension of the effects on KIM-1 immunostaining, and histopathological effect.

Response: As the reviewer suggested, we have added the images on the effects of ML-SA1 alone group in Fig. 1c, d, f, g and Fig. 3a-d.

Q7: Page 6, line 113-114: The sensitivity of the markers of function CRE and BUN should be discussed as they are known to be augmented only when high injury of the kidney occurs.

Response: Thank you for your kind suggestion. We have now discussed the implication of serum CRE and BUN in assessing the renal dysfunction and added the literature. (lines 109-110).

Q8: Page 8 and figure 1a: How do you explain that U content of the kidney is still very high and correlated to tubular damage in animals treated with ML-SA1. Justification should be added in the text.

Response: Thank you for your kind suggestion. Twenty-four hour-delayed administration of ML-SA1 significantly decreased the U content in the kidney compared to U-exposed groups alone (Fig. 1b). Simultaneously, twenty-four hour-delayed administration of ML-SA1 significantly decreased the U-induced tubular damages (KIM-1 level, serum CRE and BUN and renal pathological injuries with PT necrosis) (Fig.1c-g). We added these points in the revised manuscript (lines 201-204).

Q9: Page 9 and figure 1b: How do you explain the absence of dose-response effect of ML-SA1?

Response: We thank the reviewer for bringing up this issue. Our study found that ML-SA1 reduced the U content in the kidney through promoting lysosomal exocytosis. Absence of dose-response effect of U removal of ML-SA1 at 400 and 800 µg/kg may be due to a limitation on the number of lysosomes in which exocytosis occurs. We have

added a discussion on this issue in the revised manuscript (lines 247-249).

Q10: Page 9 and figure 1f, g: What kind of injury is scored on PT; necrosis, inflammation, fibrosis? It should be precise in the figure and in the text.

Response: Thank you for your kind suggestion. As the reviewer suggested, we have added the detailed information on PT (PT with necrotic or exfoliated cells) in the Figure legends (Fig.1g, Supplementary Fig. 1g) and in the text (line 122, line 192).

Q11: Page 12, line 254: Comparison to literature references on HK-2 cells should be cited here (Prat O 2005).

Response: As the reviewer suggested, we have now cited this reference (line 282).

Q12: Page 18, line 377: Comparison to kidney cells should be cited here (Suhard D, 2018)

Response: As the reviewer suggested, we have now cited this reference (line 482).

Q13: Page 20, line 427: What is the influence of perfusion on U tissue content? Perfusion is done with formalin?

Response: It was reported that 60–86% of U entering general circulation is excreted in the urine, and 60–70% of U is rapidly excreted in the first 24 hours while 20% remains in target organs, such as the kidneys and bones. (Wrenn et al., Radiat. Prot. Dosim. 1994, 53:255–258; Leggett RW. Health Physics. 1989, 57:365-383). In our animal experiments, kidney perfusion was performed at 48 h after single-dose U exposure or on day 6 after multiple-dose U exposure, and therefor U was mainly accumulated within the renal cells when we started the perfusion. Thus, the perfusion had a little or no effects on U tissue content. In addition, the mice need to be irrigated to obtain normal renal histomorphological observations. Perfusion was done with PBS at 4°C. We have added this information in the revised manuscript (line 532).

Q14: Materials and methods:

Page 27, TEM: How do you verify that morphology and uranium distribution is preserved by centrifugation before the addition of fixative chemicals?

Response: We thank the reviewer for bringing up this issue. Before the centrifugation at 3000 rpm, HK-2 cells were fixed with 2.5% glutaraldehyde fixative solution at room temperature for 5 min. The electron microscope images revealed that there were no changes in the morphology and distribution of organelles after the centrifugation. The mass of U crystals in the cytosol was similar to cellular organelles, therefore the centrifugation at 3000 rpm for 2 min was unlikely to cause the migration of U crystals in the cytosol.

Comments from Reviewer #3 (Remarks to the Author):

In this manuscript, Zhong et al analyze the role of TRPML1-mediated lysosomal exocytosis in the treatment of uranium-induced nephrotoxicity. They show that administration of the TRPML1 agonist ML-SA1 ameliorates the kidney phenotype of mice exposed to single or multiple doses of uranium. ML-SA1 treatment results in decreased intracellular accumulation and enhanced extracellular release of uranium, both in cellular and animal models. Mechanistically, the authors show that ML-SA1 treatment induces lysosomal exocytosis through activation of TFEB.

This is a well-written manuscript and the experimental flow is logically linked. The findings that TRPML1 activation improves uranium-induced kidney injury may be medically relevant. However, a beneficial role for TRMPL1-mediated TFEB activation has already been shown in oxalate nephropathy, in which, similarly to uranium treatment, kidney injury is observed upon oxalate-mediated induction of lysosomal damage (PMID: 32989250). Furthermore, there are a few mechanistic issues that need to be addressed. For instance, the mechanism by which uranium leads to TFEB activation is unclear and requires further elucidation. Finally, there are a few conceptual and technical points that also need to be clarified.

Specific points:

Q1. The mechanism by which uranium treatment leads to TFEB nuclear translocation is unclear. TFEB is known to be modulated by mTORC1-mediated phosphorylation, via a substrate-selective mechanism that relies on the activity of RagC/D GTPases (PMID: 32612235, 35358174). Does uranium treatment induce TFEB dephosphorylation? Does it lead to general inhibition of mTORC1 activity (i.e. impaired phosphorylation of its well-characterized substrates S6K and 4E-BP1)? Does the expression of a constitutively active RagC mutant (S75L) prevent uranium-induced TFEB nuclear translocation?

Response: We thank the reviewer for bringing up this important issue and providing constructive suggestions. We have now performed additional experiments to further dissect the mechanism underlying uranium treatment-mediated TFEB nuclear translocation (new Fig. 6 and Supplementary Fig. 8-9). We found that 24 h-U exposure alone induced the TFEB dephosphorylation (Fig. 6c, d) and inhibited the mTORC1 activity (i.e., the inhibition of phosphorylation of S6 kinase) (Fig. 6k, l), and that ML-SA1 had no effect on mTORC1 activity in HK-2 cells (Fig. 6k, l). Moreover, overexpression of constitutively active RagC GTPases completely abolished U-induced TFEB nuclear translocation and inhibited the TFEB nuclear translocation induced by Torin1, a strong mTOR inhibitor (Supplementary Fig. 9e, f), indicating that U exposure-induced TFEB nuclear translocation was mainly mediated by inhibiting the mTORC1 activity via a substrate-selective mechanism that relies on the activity of RagC/D GTPases, which is associated with the specific effect of lysosomal damage and subsequent TRPML1 activation on RagC/D activity⁶¹. We also found that U-induced TFEB nuclear translocation was impaired in the HK-2 cells with knockdown of TRPML1 or PPP3CB, but still obviously higher than that of vector- or scramble siRNA-transfected cells (Fig. 6i, j, Supplementary Fig. 8a-f), suggesting that U-induced TFEB nuclear translocation was partly through TRPML1 and calcineurin and additional mechanisms may be involved, i.e. inhibiting the mTORC1 activity via a substrate-selective mechanism that relies on the activity of RagC GTPases.

Q2. Related to the previous point and as stated above, a beneficial role for TRMPL1-

mediated TFEB activation has already been shown in oxalate nephropathy (PMID: 32989250). This reference should be added and discussed by the authors. Interestingly, in this model oxalate-induced lysosomal damage was shown to induce TFEB activation through a TRPML1-mediated mechanism that modulates the activity of RagC/D GTPases. Thus, it would be important to assess if uranium promotes TFEB activity through a similar mechanism.

Response: As described in the response of Q1 of the reviewer, we have performed additional experiments and demonstrated that 24 h-U exposure alone impaired the mTORC1-mediated phosphorylation of TFEB in a RagC GTPase-dependent manner, suggesting that lysosomal damage and TRPML1 activation specifically regulate RagC activity, leading to the dephosphorylation of TFEB. Our results are consistent with the recent report showing that LLOMe-induced lysosomal damage induced TFEB activation through the TRPML1-mediated Ca^{2+} efflux from lysosomes and consequent the activity of RagC/D GTPases⁶¹. This reference has been added and discussed in the text (lines 389-403).

Q3. The authors show that uranium induces TFEB nuclear translocation, which then leads to TRPML1-mediated lysosomal exocytosis. They also show that TRPML1 activation by ML-SA1 strongly enhances lysosomal exocytosis in HK2 cells in a TFEB-dependent manner, as TFEB depletion blunts ML-SA1-induced lysosomal exocytosis. However, TRPML1 is the major effector of TFEB-induced lysosomal exocytosis and should be functionally downstream of TFEB. How do the authors explain this discrepancy? Are the expression levels of TRPML1, a direct TFEB transcriptional target, too low in TFEB-depleted cells for proper activation by ML-SA1? Does TRPML1 overexpression promote lysosomal exocytosis, and does it protect from uranium-induced cell death, both in control and TFEB-depleted HK-2 cells?

Response: We thank the reviewer for bringing up these issues. It has been reported that TFEB promotes lysosomal exocytosis through the activation of the lysosomal Ca^{2+} channel MCOLN1⁵⁰. We demonstrated that U induced the TFEB nuclear translocation and then regulated the TRPML1-mediated lysosomal exocytosis as knockdown of

TFEB decreased the U-induced and ML-SA1-activated extracellular β -hex content (Fig. 7c). TRPML1 activated by exogenous ligand ML-SA1 also promoted the TFEB nuclear translocation through TRPML1-mediated calcineurin activation (Fig. 6i, j). The effects of TFEB knockdown on abolishing ML-SA1-activated lysosomal exocytosis attribute to a very low level of TRPML1, resulting from the positive feedback loop between TRPML1 and TFEB in TFEB-depleted cells (lines 437-439). We repeated the experiments in Fig. 7c, e, f, and the results showed that ML-SA1 didn't increase the extracellular β -hex content and decrease the cell death in TFEB-depleted cells exposed to U (Fig. 7 c, f).

As shown in Fig. 6a, b, e, f, g, h, exogenous ligand ML-SA1 activated the lysosomal Ca^{2+} release, TFEB nuclear translocation and TRPML1 expression in both control and U-loaded HK-2 cells. ML-SA1-induced nuclear translocation of TFEB in both control and U-loaded HK-2 cells were abolished by shRNA-mediated depletion of TRPML1 and siRNA-mediated depletion of PPP3CB, one of the subunits of calcineurin, respectively (Fig. 6i, j, Supplementary Fig. 8a-f). However, ML-SA1 didn't promote lysosomal exocytosis and protect against U-induced cell death in TFEB-depleted HK-2 cells (Fig. 7c, f), which attributed to very low expression levels of TRPML1 (a direct TFEB transcriptional target) in TFEB-depleted cells. These data suggested that TRPML1 overexpression can promote the lysosomal exocytosis and protect from U-induced cell death both in control and TFEB-depleted HK-2 cells.

Q4. Related to point 3, is the activation of TFEB, using TRPML1-unrelated drugs (e.g. torin treatment), sufficient to promote lysosomal exocytosis and protection from cell death in uranium-treated HK2 cells?

Response: We thank the reviewer for bringing up these issues and constructive suggestions. We found that Torin 1, mTORC1 inhibitor, promoted lysosomal exocytosis (i.e., increased the extracellular β -hex content and U content and decreased the intracellular U content) and protected against cell death in U-loaded HK-2 cells (data not shown). TFEB activation could be another approach to clear the U in the kidney, which is worthy of further study in the near future.

Q5. The analysis of lysosomal exocytosis in cellular systems has been only done by assessing the secretion of lysosomal enzymes. Additional assays are required for proper characterization of lysosomal exocytosis. For instance, plasma membrane localization of LAMP1, using antibodies recognizing the luminal region of the protein, should be also determined by FACS and immunofluorescence analysis in non-permeabilized cells subjected to uranium and ML-SA1 treatment.

Response: As suggested, we performed the immunofluorescence analyses for the LAMP-1 luminal domain in non-permeabilized cells and total TRPML1 in permeabilized cells subjected to uranium and ML-SA1 treatment. We found that 24 h-delayed ML-SA1 treatment enhanced the exposure of luminal domain of LAMP-1 on the PM in both control cells and U-loaded HK-2 cells at 30 min after 24 h-U exposure at 600 μ M (Fig. 4b). Moreover, 24 h-delayed ML-SA1 treatment induced the TRPML1 localization around the cells in both control cells and U-loaded HK-2 cells at 30 min after 24 h-U exposure at 600 μ M (Fig. 4b), suggesting a PM localization of the protein (line 301).

Q6. Higher quality images for immunofluorescence analyses in cell lines are needed. For instance, in figure 5a cells are too small and can be barely seen. Furthermore, the fluorescence signal is heavily saturated. Thus, no conclusions can be drawn from co-localization analyses using these images. A similar issue also occurs for TFEB localization shown in Figure 6. Higher resolution and higher magnification images need to be provided, using non-saturated signals. Co-localization analyses should be then performed using the newly acquired images only.

Response: We thank the reviewer for bringing up this issue, which is similar to Q5 raised by the reviewer 1. We have repeated the experiments described in Figure 5a, d and found that supersaturation of the fluorescent signal led to low-quality of images. We re-performed the colocalization experiments of galectin-3 and LAMP-1 and colocalization analyses were re-performed using the newly acquired images. We also repeated the experiments for TFEB localization shown in Figure 6e, f. We re-analyzed

the TFEB nuclear translocation with the newly acquired images.

Q7. The number of cells used for the quantifications of cellular assays need to be provided, together with detailed information on the statistical analyses used.

Response: As suggested, we have added the information of the number of cells used for the quantifications of cellular assays of immunofluorescence staining (line 639) and Calcein-AM/PI staining (line 646). We also added the detailed information on the statistical analyses (lines 688-689), and re-checked the results of statistical analyses and corrected the faults.

REVIEWER COMMENTS

Reviewer #1 (Remarks to the Author):

The revision is satisfying and the manuscript can be accepted.

Reviewer #2 (Remarks to the Author):

The authors responded appropriately to the majority of comments and completed the manuscript with interesting additional data.

Nevertheless, there are some elements to correct or complete:

- Page 23, line 482: Ref 14 on kidney cells and not on neurons as mentioned in the sentence
- Q13: The sentence "Thus, the perfusion had a little or no effects on U tissue content" requires being documented by experiences or references
- Q14 and Supplementary Data 11: The influence of centrifugation on cell morphology is usually observed in cellular studies, how do the authors demonstrate that this is not the case? The methodology for preparing samples for TEM must be specified (fixing of samples, contrast agent, cutting thickness, identification of the uranium spectrum ...)
-

Reviewer #3 (Remarks to the Author):

The authors addressed most of my previous concerns. However, there are some issues that still need to be addressed:

1. The analysis and interpretation of mTORC1/TFEB activation by uranium (U) is confusing. As the authors state, TFEB localization and activity are controlled by RagC/D-dependent mTORC1-mediated phosphorylation. The evidence shown here (Suppl. Fig. 9e, f) that overexpression of constitutively active RagC rescues TFEB subcellular localization in Torin-treated cells indicates that the expression levels of active RagC in these experimental settings are extremely high and lead to a sequestration artifact of TFEB. Indeed, the mechanism by which active RagC is known to regulate TFEB localization is by promoting mTORC1-mediated phosphorylation, which however cannot occur in the presence of Torin1, a potent mTORC1 inhibitor. Therefore, the data provided in the presence of RagC likely reflect an overexpression artifact and should be performed using proper RagC amounts. Please also note that images in Suppl. Fig. 9 are of low quality and no proper conclusion can be drawn by looking at these data.

Furthermore, as RagC/D activity is mainly important for TFEB phosphorylation but does not affect the phosphorylation of other mTORC1 substrates (e.g. S6K), the data in Fig 6k showing inhibition of S6K phosphorylation at high levels (600 μ M), but not at low levels (100 μ M) of U treatment, indicate that uranium may lead to specific RagC/D inactivation at low levels only, whereas high U concentration likely causes general cell toxicity thus leading to mTORC1 inhibition on all substrates. This should be clearly stated and discussed to avoid confusion in the interpretation of the data.

Finally, the analysis of TFEB phosphorylation in Fig. 6c should be performed with higher quality experiments. Due to the low expression of TFEB in most cell lines (including HK-2 cells), the use of phospho-antibodies to analyze endogenous TFEB is unreliable. No shift in the molecular weight of TFEB, which is used as a reliable method to analyze the phosphorylation of endogenous TFEB, is observed in Fig. 6c. This is a bit controversial and suggests that the authors did not use the proper electrophoresis conditions to observe such a shift.

2. The authors state that ML-SA1 and U treatment "significantly induced LAMP-1 and TRPML1 localization to the apical membrane in S1, S2 and S3 segments of proximal tubules (Fig. 2a, b and Supplementary Fig. 5)". However, in Fig 2 both LAMP1 and TRPLM1 show apical localization even in control sections, and no difference is visible between control and treated samples. The quantifications provided in Fig 2b refer to either "No of LAMP1-positive PT/field" or to "Intensity of TRPML1-positive PT/field", which only reflect the quantity, rather than localization, of lysosomes. In Suppl. Fig. 5,

images are of low quality and no magnification or quantification is provided. Thus, the authors' conclusion that ML-SA1 and U treatment promotes lysosomal exocytosis in vivo is not supported by their data and no proper quantification analyses are provided.

3. Despite the authors' attempts to perform a new analysis of galectin3 co-localization with LAMP1, the new images still look of poor quality, and analyses to calculate galectin3/LAMP1 co-localization have not been performed properly. The authors show in Supplementary Fig. 7h the "percentage of cells with colocalization of galectin3/LAMP1". However, this analysis looks highly arbitrary as it is difficult to determine and define "cells with colocalization" from cells "without" colocalization. Standard co-localization analyses (e.g. Manders' colocalization coefficient) should be performed in any co-localization experiment. Furthermore, the number of cells analyzed, which has been added during the revision by the authors in the methods as a general number of cells analyzed in all the experiments of the manuscript, should be instead provided in each figure legend throughout the manuscript, in order to indicate the number of cells effectively analyzed in each experiment.

4. Lane 386. The authors state: "Studies have revealed that TFEB nuclear export is mainly negatively modulated by mTORC1 (mechanistic target of rapamycin complex 1)-mediated TFEB phosphorylation via a substrate-selective mechanism that relies on the activity of RagC/D GTPases 81-88". These studies have demonstrated that not only TFEB nuclear export, but also import, is controlled by mTORC1-mediated TFEB phosphorylation. I suggest rephrasing the sentence in this way: "Studies have revealed that TFEB nucleo-cytoplasmic shuttling is mainly modulated by..."

Response to Reviewers

Comments from Reviewer #2 (Remarks to the Author):

The authors responded appropriately to the majority of comments and completed the manuscript with interesting additional data.

Nevertheless, there are some elements to correct or complete:

Q1. -Page 23, line 482: Ref 14 on kidney cells and not on neurons as mentioned in the sentence.

Response: We thank the reviewer for pointing this out. We have corrected the error by deleting the Ref 14 in the sentence (page 23, line 490).

Q2. -Q13: The sentence "Thus, the perfusion had a little or no effects on U tissue content" requires being documented by experiences or references.

Response: We thank the reviewer for bringing up this issue. As suggested, we have performed the experiment and found that the perfusion had no effects on U content in kidney of mice with a single dose of U at 0.4 mg/kg and 2.0 mg/kg body weight and multiple doses of U at 80 µg/kg body weight once daily for 5 days (page 27, lines 572-575, supplementary Table 1).

Q3. -Q14 and Supplementary Data 11: The influence of centrifugation on cell morphology is usually observed in cellular studies, how do the authors demonstrate that this is not the case? The methodology for preparing samples for TEM must be specified (fixing of samples, contrast agent, cutting thickness, identification of the uranium spectrum ...).

Response: We agree that centrifugation of the sample before fixation may cause some distortion to the cellular contents, especially when there is not a well-defined speed to centrifuge cells into a pellet for TEM process. On the other side, it has been reported that centrifugation under certain speeds yielded good results in TEM studies. For examples, Graham et al. centrifuged at 9,500 g for 5-10 min at room temperature (Graham et al. Nature protocols, 2007, 2(10):2439-2450), Nativo et al. centrifuged at

5,000 g for 5 min (Nativo, P., et al. ACS Nano, 2008, 2, 1639-1644), and Schrand et al. centrifuged at 1,000 g for 5 min (Schrand et al. Nature protocols, 2010, 5(4):744-757). The determination of centrifugal speed and time may be related to a variety of factors such as *in situ* fixation time. In our preparation of cells for TEM, HK-2 cells were scraped after 5 min of *in situ* fixation with 2.5% glutaraldehyde fixative solution at room temperature, pelleted the cells in 15 ml centrifuge tube at 3,000 rpm (convert rpm to g, 1158 g) for 2 min, and re-fixed with 2.5% glutaraldehyde fixative solution at room temperature for 30 min. We compared the cell morphology in our TEM images of HK-2 cells with those in TEM images of various cells in the literature including cultured pig renal proximal tubule epithelial cells (LLC-PK1 cells)²⁴, rat renal proximal tubule epithelial cells (NRK-52^E cells)²², osteoblastic cells (ROS 17/2.8²¹ and UMR-106²⁰). It was found that there were no marked differences in morphology of cell membrane, nucleus, mitochondria or endoplasmic reticulum between our TEM images of HK-2 cells and TEM images in the literature^{20, 21, 22, 24}.

We further tested whether uranyl acetate and lead citrate staining had an effect on the formation of U precipitation inside lysosome-like vesicles. Previous studies have demonstrated that intracellular U in precipitated form displayed a characteristic electron-dense needlelike structures (like urchin-shaped precipitates) mainly located in the lysosomes, multivesicular bodies or autophagic vesicles, which contain U and phosphorus elements identified by EDS, EDX and EXAFS analysis in cultured LLC-PK1 cells²⁴, NRK-52^E cells^{22, 23}, ROS 17/2.8 cells²¹, UMR-106 cells²⁰ and human cerebral microvascular endothelial cells (hCMEC/D3)¹⁹. Moreover, there was no difference in the formation of needle-shaped U precipitates inside lysosome-like vesicles in TEM images of ultra-thin sections stained with^{20, 24}/without^{19, 21, 22, 23} uranyl acetate and/or lead citrate. Morphologically, our TEM analyses showed the formation of U precipitates with electron-dense needle-like structure in lysosomes and multivesicular bodies, which were grown in a U concentration-dependent manner in HK-2 cells after 24 h exposure to U at 50, 100, 300 and 600 μ M (Supplementary Fig. 11a). In contrast, HK-2 cells without U exposure did not show the electron-dense

needle-shaped structure inside the lysosomes by TEM (we re-prepared the TEM sample for HK-2 cells without exposure to U using the same protocol for preparing the TEM samples and new TEM image of control HK-2 cells showed in the first column of Supplementary Fig. 11a). Moreover, no needlelike U precipitates inside lysosome-like vesicles were observed by TEM in HK-2 cells with U exposure at 10 μ M for 10, 20 and 30 d (Supplementary Fig. 11b). In addition, although we were unable to conduct the experiment to identify the U spectrum in TEM samples, U content in the U-exposed HK-2 cells was significantly increased by ICP-MS (Supplementary Fig. 7b). These results suggested that the formation of U precipitates with electron-dense needle-like structure inside lysosome-like vesicles was not caused by the uranyl acetate staining in the process of sample preparation for TEM.

As reviewer suggested, we have added the methodology for preparing samples for TEM including contrast agent (uranyl acetate and lead citrate), cutting thickness (60-80 nm) in Supplementary Materials and Methods.

Comments from Reviewer #3 (Remarks to the Author):

The authors addressed most of my previous concerns. However, there are some issues that still need to be addressed:

Q1. The analysis and interpretation of mTORC1/TFEB activation by uranium (U) is confusing. As the authors state, TFEB localization and activity are controlled by RagC/D-dependent mTORC1-mediated phosphorylation. The evidence shown here (Suppl. Fig. 9e, f) that overexpression of constitutively active RagC rescues TFEB subcellular localization in Torin-treated cells indicates that the expression levels of active RagC in these experimental settings are extremely high and lead to a sequestration artifact of TFEB. Indeed, the mechanism by which active RagC is known to regulate TFEB localization is by promoting mTORC1-mediated phosphorylation, which however cannot occur in the presence of Torin1, a potent mTORC1 inhibitor. Therefore, the data provided in the presence of RagC likely reflect an overexpression artifact and should be performed using proper RagC amounts. Please also note that

images in Suppl. Fig. 9 are of low quality and no proper conclusion can be drawn by looking at these data.

Furthermore, as RagC/D activity is mainly important for TFEB phosphorylation but does not affect the phosphorylation of other mTORC1 substrates (e.g. S6K), the data in Fig 6k showing inhibition of S6K phosphorylation at high levels (600 μ M), but not at low levels (100 μ M) of U treatment, indicate that uranium may lead to specific RagC/D inactivation at low levels only, whereas high U concentration likely causes general cell toxicity thus leading to mTORC1 inhibition on all substrates. This should be clearly stated and discussed to avoid confusion in the interpretation of the data.

Finally, the analysis of TFEB phosphorylation in Fig. 6c should be performed with higher quality experiments. Due to the low expression of TFEB in most cell lines (including HK-2 cells), the use of phospho-antibodies to analyze endogenous TFEB is unreliable. No shift in the molecular weight of TFEB, which is used as a reliable method to analyze the phosphorylation of endogenous TFEB, is observed in Fig. 6c. This is a bit controversial and suggests that the authors did not use the proper electrophoresis conditions to observe such a shift.

Response: We thank the reviewer for bringing up these issues. We re-performed the experiment on effect of overexpression of constitutively active RagC GTPase on U-induced TFEB nuclear translocation by transfection cell with proper RagC amounts, and images in Supplementary Fig. 9 re-captured and quality of images have improved. The results showed that overexpression of constitutively active RagC GTPases significantly inhibited TFEB nuclear translocation induced by 24 h-U exposure at 100 and 600 μ M and this inhibition was again reverted by concomitant treatment with mTORC1 inhibitor Torin1 (Supplementary Fig. 9e, f), indicating that 24 h-U exposure alone impairs the mTORC1-mediated phosphorylation of TFEB by inactivating RagC GTPase (pages 19-20, lines 402-406).

We repeated the experiments in Fig. 6k, l. The data showed the inhibition of S6K phosphorylation at high levels (600 μ M), but not at low levels (100 μ M) of U exposure. Combined with effect of U exposure at 100 and 600 μ M on inactivating RagC GTPase,

it can be concluded that U exposure led to specific RagC/D inactivation at both low and high levels, whereas only high U concentration led to the inhibition of mTORC1-mediated phosphorylation of TFEB and S6K, which may be due to the general cellular toxicity caused by U at the high concentration (page 19, lines 392-399). Although RagC is dispensable for mTORC1-mediated S6K phosphorylation, it has been reported that overexpression of RagC mutants inhibits S6K phosphorylation likely in a dominant-negative fashion⁹⁰, suggesting that U-exposure at the high concentration (600 μ M) may also inhibit mTORC1 activity by inducing RagC mutant activity in a dominant negative pattern. Therefore, U-induced impairment of the recruitment of mTORC1 and TFEB to lysosomes, which could be caused by the RagC inactivation, mTORC1 inhibition and/or dominant negative RagC mutant, mediates the transcriptional activation of TRPML1, leading to U-induced TFEB activation upon nuclear translocation (page 20, lines 408-415).

We re-performed the Western blotting analysis of endogenous TFEB phosphorylation using anti-TFEB antibody and the proper electrophoresis conditions to monitor the shift in the molecular weight of TFEB. As expected, ML-SA1 treatment significantly increased the dephosphorylation of TFEB (TFEB molecular weight downshift) in both control and U-loaded HK-2 cells with 24-h exposure at 100 and 600 μ M (Fig. 6c, d) (page 17, lines 351-353), which was consistent with the effects of ML-SA1 on an increase of nuclear translocation of TFEB and levels of the TFEB downstream targets LAMP-1 and TRPML1 proteins in both control and U-loaded HK-2 cells with 24-h exposure at 100 and 600 μ M (Fig. 6e-h).

Q2. The authors state that ML-SA1 and U treatment “significantly induced LAMP-1 and TRPML1 localization to the apical membrane in S1, S2 and S3 segments of proximal tubules (Fig. 2a, b and Supplementary Fig. 5)”. However, in Fig 2 both LAMP1 and TRPLM1 show apical localization even in control sections, and no difference is visible between control and treated samples. The quantifications provided in Fig 2b refer to either “No of LAMP1-positive PT/field” or to “Intensity of TRPML1-

positive PT/field”, which only reflect the quantity, rather than localization, of lysosomes. In Suppl. Fig. 5, images are of low quality and no magnification or quantification is provided. Thus, the authors’ conclusion that ML-SA1 and U treatment promotes lysosomal exocytosis in vivo is not supported by their data and no proper quantification analyses are provided.

Response: This is also a valuable concern and we thank the reviewer for raising it, giving us the opportunity to clarify the issue. As reviewer mentioned, both LAMP-1 and TRPLM1 show apical localization in control and treated samples. Polarized exocytosis of lysosomes can be monitored by the asymmetric appearance of LAMP-1 or LAMP-2 on the PM in polarized epithelial cells^{68,69}. Studies have revealed that before lysosomes fuse with PM, lysosomes translocate from peri-nuclear zone to the region adjacent to PM^{30,33,50,67}. We found that LAMP-1 as well as TRPML1 was localized to the apical membrane, but not to the basolateral membrane, of S1, S2 and S3 segments of proximal tubules of kidney in control mice (Fig. 2a, b and Supplementary Fig. 5), indicating that lysosomal exocytosis directed toward the proximal tubular lumen is active at the normal renal physiological conditions (page 11, lines 225-229). Importantly, quantity difference in “No of LAMP1-positive PT/field” and “Intensity of TRPML1-positive PT/field” among control and treated groups reflect the extent of lysosomal exocytosis.

We improved the quality of images in Supplementary Fig. 5. Scale bar in 10 μm was too short to be seen clearly. We increased the scale bar to 20 μm in Supplementary Fig. 5 and other figures (Figs. 1-3, Supplementary Figs. 1-4, 6). The images in Fig. 2a were images of S3 or S1 segments of proximal tubules in control and treated groups, and the images in Supplementary Fig. 5 were images of S1 and S2 or S2 and S3 segments of proximal tubules in control and treated groups, their quantification were shown in Fig. 2b. In addition, we included the quantitative analysis of LAMP-1 and TRPML1 staining of the apical membrane in the S1, S2 and S3 segments of renal proximal tubules by adding three mice in the experiments on multiple-dose U exposure (80 $\mu\text{g}/\text{kg}/\text{day}$ for 5 days) followed by ML-SA1 treatment (column 1 to the right of Fig.

2b, line 968), which further suggests that ML-SA1 promotes the apical exocytosis of lysosomes in renal PTECs of mice exposed to multiple low-dose U.

Q3. Despite the authors' attempts to perform a new analysis of galectin3 co-localization with LAMP1, the new images still look of poor quality, and analyses to calculate galectin3/LAMP1 co-localization have not been performed properly. The authors show in Supplementary Fig. 7h the "percentage of cells with colocalization of galectin3/LAMP1". However, this analysis looks highly arbitrary as it is difficult to determine and define "cells with colocalization" from cells "without" colocalization. Standard co-localization analyses (e.g. Manders' colocalization coefficient) should be performed in any co-localization experiment. Furthermore, the number of cells analyzed, which has been added during the revision by the authors in the methods as a general number of cells analyzed in all the experiments of the manuscript, should be instead provided in each figure legend throughout the manuscript, in order to indicate the number of cells effectively analyzed in each experiment.

Response: We thank the reviewer for bringing up this issue. We repeated the colocalization experiments of galectin-3 and LAMP-1 described in Fig. 5a, d. We re-captured high-resolution images to improve the quality of images and analysis of galectin-3 colocalization with LAMP1 were re-performed using the newly acquired images using Manders' colocalization coefficient (New Fig. 5a, d).

As reviewer suggested, we added the number of cells analyzed in each figure legend for quantitative analysis involving the number of cells throughout the manuscript [Figure legends for Fig. 5 (lines 1006-1010), Fig. 6f, i, j (line 1032), Supplementary Fig. 7f, h, Supplementary Fig. 9b, d, f].

Q4. Lane 386. The authors state: "Studies have revealed that TFEB nuclear export is mainly negatively modulated by mTORC1 (mechanistic target of rapamycin complex 1)-mediated TFEB phosphorylation via a substrate-selective mechanism that relies on the activity of RagC/D GTPases 81-88". These studies have demonstrated that not only

TFEB nuclear export, but also import, is controlled by mTORC1-mediated TFEB phosphorylation. I suggest rephrasing the sentence in this way: “Studies have revealed that TFEB nucleo-cytoplasmic shuttling is mainly modulated by...”.

Response: We thank the reviewer for bringing up this issue and providing constructive suggestions. We have rephrased the sentence as reviewer suggestion (line 387).

REVIEWERS' COMMENTS

Reviewer #3 (Remarks to the Author):

The authors addressed all my previous concerns, and I am now satisfied with their revision. There is only one final issue that concerns the interpretation of RagC data. The authors state that “Although RagC is dispensable for mTORC1-mediated S6K phosphorylation, it has been reported that overexpression of RagC mutants inhibits S6K phosphorylation likely in a dominant negative fashion⁹⁰, suggesting that U-exposure at the high concentration (600 μ M) may also inhibit mTORC1 activity by inducing RagC mutant activity in a dominant negative pattern”. I do not understand what the authors mean with this interpretation. The authors show that uranium (U) exposure at high concentration promotes S6K de-phosphorylation in wt, non-transfected, HK2 cells (Fig 6K). How could mutant RagC lead to a dominant negative effect, if mutant RagC is not even expressed in these cells? There is now robust evidence showing that RagC activity is dispensable for S6K phosphorylation (Lawrence et al., *Science* 2019; Napolitano et al., *Nature* 2020; Nakamura et al., *Nat Cell Biol* 2020; Goodwin et al., *Sci Advances* 2021; Li et al., *Plos Biol* 2022). Thus, the inhibitory effect on S6K, observed at high U concentration, is likely due to toxicity-induced inhibition of RagA/B, which are major modulators of S6K phosphorylation. While at this stage of revision I would not ask the authors to make experiments to demonstrate if this hypothesis is correct, I believe they should at least acknowledge this possibility in their manuscript and remove the hypothesis that mutant RagC may play a dominant negative effect. As discussed above, this cannot be an issue in the proposed model.

Response to Reviewers

Comments from Reviewer #3 (Remarks to the Author):

The authors addressed all my previous concerns, and I am now satisfied with their revision.

Q. There is only one final issue that concerns the interpretation of RagC data. The authors state that “Although RagC is dispensable for mTORC1-mediated S6K phosphorylation, it has been reported that overexpression of RagC mutants inhibits S6K phosphorylation likely in a dominant negative fashion⁹⁰, suggesting that U-exposure at the high concentration (600 μ M) may also inhibit mTORC1 activity by inducing RagC mutant activity in a dominant negative pattern”. I do not understand what the authors mean with this interpretation. The authors show that uranium (U) exposure at high concentration promotes S6K de-phosphorylation in wt, non-transfected, HK2 cells (Fig 6K). How could mutant RagC lead to a dominant negative effect, if mutant RagC is not even expressed in these cells?

There is now robust evidence showing that RagC activity is dispensable for S6K phosphorylation (Lawrence et al., Science 2019; Napolitano et al., Nature 2020; Nakamura et al., Nat Cell Biol 2020; Goodwin et al., Sci Advances 2021; Li et al., Plos Biol 2022). Thus, the inhibitory effect on S6K, observed at high U concentration, is likely due to toxicity-induced inhibition of RagA/B, which are major modulators of S6K phosphorylation. While at this stage of revision I would not ask the authors to make experiments to demonstrate if this hypothesis is correct, I believe they should at least acknowledge this possibility in their manuscript and remove the hypothesis that mutant RagC may play a dominant negative effect. As discussed above, this cannot be an issue in the proposed model.

Response: We thank the reviewer for bringing up this issue and providing constructive suggestions. As suggested, we removed the discussion on dominant negative effect of mutant RagC, and added the hypothesis that the inhibitory effect on S6K caused by U-exposure at the high concentration is likely due to toxicity-induced inhibition of RagA/B.